# An essential role for *Argonaute 2* in EGFR-KRAS signaling in pancreatic cancer development

Sunita Shankar[1,2,9], Jean Ching-Yi Tien[1,2,9], Ronald F. Siebenaler [1,2,9], Seema Chugh[1,2,9], Vijaya L. Dommeti[1,2], Sylvia Zelenka-Wang[1,2], Xiao-Ming Wang[1,2], Ingrid J. Apel[1,2], Jessica Waninger[1,2], Sanjana Eyunni[1,2], Alice Xu[1,2], Malay Mody[1,2], Andrew Goodrum[1,2], Yuping Zhang[1,2], John J. Tesmer[3], Rahul Mannan[1,2], Xuhong Cao[1,2,4], Pankaj Vats [1,2], Sethuramasundaram Pitchiaya [1,2], Stephanie J. Ellison[1,2], Jiaqi Shi[2], Chandan Kumar-Sinha[1,2], Howard C. Crawford[5,6] & Arul M. Chinnaiyan [1,2,4,7,8✉]

Both KRAS and EGFR are essential mediators of pancreatic cancer development and interact with Argonaute 2 (AGO2) to perturb its function. Here, in a mouse model of mutant KRAS-driven pancreatic cancer, loss of *AGO2* allows precursor lesion (PanIN) formation yet prevents progression to pancreatic ductal adenocarcinoma (PDAC). Precursor lesions with *AGO2* ablation undergo oncogene-induced senescence with altered microRNA expression and EGFR/RAS signaling, bypassed by loss of *p53*. In mouse and human pancreatic tissues, PDAC progression is associated with increased plasma membrane localization of RAS/AGO2. Furthermore, phosphorylation of $AGO2^{Y393}$ disrupts both the wild-type and oncogenic KRAS-AGO2 interaction, albeit under different conditions. ARS-1620 (G12C-specific inhibitor) disrupts the $KRAS^{G12C}$-AGO2 interaction, suggesting that the interaction is targetable. Altogether, our study supports a biphasic model of pancreatic cancer development: an *AGO2*-independent early phase of PanIN formation reliant on EGFR-RAS signaling, and an *AGO2*-dependent phase wherein the mutant KRAS-AGO2 interaction is critical for PDAC progression.

[1] Michigan Center for Translational Pathology, University of Michigan, Ann Arbor, MI 48109, USA. [2] Department of Pathology, University of Michigan, Ann Arbor, MI 48109, USA. [3] Department of Biological Sciences, Purdue University, West Lafayette, IN 47907, USA. [4] Howard Hughes Medical Institute, University of Michigan, Ann Arbor, MI 48109, USA. [5] Department of Molecular and Integrative Physiology, University of Michigan, Ann Arbor, MI 48109, USA. [6] Internal Medicine, University of Michigan, Ann Arbor, MI 48109, USA. [7] Department of Urology, University of Michigan, Ann Arbor, MI 48109, USA. [8] Rogel Cancer Center, University of Michigan, Ann Arbor, MI 48109, USA. [9] These authors contributed equally: Sunita Shankar, Jean Ching-Yi Tien, Ronald F. Siebenaler, Seema Chugh. ✉email: arul@umich.edu

**K**RAS mutations drive over 90% of pancreatic cancer, a disease with a dismal overall 5-year survival rate of only 9%[1]. Like all RAS GTPases, KRAS is a molecular switch that transduces extracellular mitogenic signals by cycling between an active GTP-bound and an inactive GDP-bound state. Proteins that regulate the nucleotide loading of RAS, like GTPase activating proteins (GAPs) or guanine exchange factors (GEFs), recruit RAS to the plasma membrane in response to activated growth factor receptors, such as EGFR[2,3]. Recurrent oncogenic driver mutations in *RAS* result in the accumulation of its active GTP-bound form at the plasma membrane, leading to aberrant signaling[2,3].

Genetically engineered mouse models (GEMMs) of pancreatic cancer were developed by expression of a single oncogenic $KRAS^{G12D}$ allele in the mouse exocrine pancreas. In this model, pre-invasive pancreatic intraepithelial (PanINs) lesions progress to pancreatic adenocarcinoma (PDAC) reflective of the human disease[4]. Use of such GEMMs has been instrumental in defining the key events that characterize PanIN development and PDAC progression[5,6]. Of particular relevance is the observation that EGFR is essential for $KRAS^{G12D}$-driven PanIN development[7,8]. However, the requirement for EGFR at the early stage of PanIN development has not translated to successful treatment[9], while directly targeting KRAS also remains a challenge[10].

Earlier, we identified a direct interaction between KRAS and Argonaute 2 (AGO2), independent of *KRAS* mutation status[11], which was required for oncogenic *KRAS*-driven cellular transformation. Interestingly, Shen et al. had previously shown that EGFR phosphorylates AGO2 at tyrosine 393 under hypoxic stress[12]. Here, we employed established mouse models of pancreatic cancer to determine the in vivo requirement of *AGO2* in pancreatic cancer development. Our data show that oncogenic *KRAS*-initiated PanIN formation is reliant on EGFR and wild-type RAS signaling, independent of *AGO2*. Strikingly, however, we identify a critical dependence on *AGO2* for PanIN progression to PDAC, bypassed by loss of *p53*. While defining an essential role for *AGO2* in PDAC progression, we also further our understanding of how the KRAS-AGO2 interaction is regulated through EGFR activation. Disruption of the oncogenic KRAS-AGO2 association may, therefore, represent a point of therapeutic intervention to prevent pancreatic cancer progression.

## Results

### AGO2 loss allows pancreas development and PanIN formation.
To investigate the role of *AGO2* in the development of pancreatic cancer in vivo, we employed the GEMM of pancreatic cancer initiated by a conditionally activated allele of *KRAS*[4], $KRAS^{LSL-G12D/+}$ ($KRAS^{G12D}$, Fig. 1a). Crossing $KRAS^{G12D}$ mice with animals harboring *Cre* recombinase knocked into the pancreas-specific promoter, *p48* (*p48Cre*), yields $KRAS^{G12D}$; *p48Cre* mice that develop pancreatic intraepithelial neoplasia (PanINs) precursor lesions beginning around 8 weeks[4]. Over time, these PanINs progress to pancreatic ductal adenocarcinoma (PDAC) and develop metastases. Next, we generated transgenic mice with both $KRAS^{G12D}$ and conditionally deleted allele(s) of *AGO2* (ref. [13]) (Fig. 1a). The resulting $KRAS^{G12D}$;*p48Cre* mice were either wild-type, heterozygous, or homozygous for the conditional allele of *AGO2* (hereafter referred to as $AGO2^{+/+}$;$KRAS^{G12D}$;*p48Cre*, $AGO2^{fl/+}$;$KRAS^{G12D}$;*p48Cre*, and $AGO2^{fl/fl}$;$KRAS^{G12D}$;*p48Cre*, respectively). Genomic PCR confirmed Cre-driven excision and recombination of the oncogenic *KRAS* allele[4] in pancreata from mice with $KRAS^{G12D}$; *p48Cre* alleles (Supplementary Fig. 1a). Further, qRT-PCR analysis showed significant reduction in *AGO2* expression in $AGO2^{fl/fl}$;$KRAS^{G12D}$;*p48Cre* mice (Supplementary Fig. 1b).

Histology of pancreata from mice with Cre-mediated *AGO2* ablation ($AGO2^{fl/fl}$; *p48Cre*) showed normal morphology (Fig. 1b, left panels) with no differences in pancreatic weight compared with pancreata from $AGO2^{+/+}$;*p48Cre* mice (Supplementary Fig. 1c). This suggests that loss of *AGO2* does not grossly interfere with pancreas development. Immunohistochemistry (IHC) with a monoclonal antibody specific to AGO2 (Supplementary Fig. 2, Supplementary Table 1) showed minimal expression of AGO2 in the acinar cells of both $AGO2^{+/+}$;*p48Cre* and $AGO2^{fl/fl}$;*p48Cre* pancreata (Fig. 1b, right panels). These data indicate a non-essential role for *AGO2* in the acinar cells during normal pancreatic development. However, expression of $KRAS^{G12D}$ in the pancreatic acinar cells led to increased AGO2 expression in the PanINs as well as the surrounding stroma in 12-week-old $AGO2^{+/+}$;$KRAS^{G12D}$;*p48Cre* mice (Fig. 1c, top panels). Notably, we observed PanIN lesions in $AGO2^{fl/fl}$;$KRAS^{G12D}$;*p48Cre* pancreata lacking *AGO2* expression (Fig. 1c, lower panels) that were morphologically indistinguishable from those arising in $AGO2^{+/+}$;$KRAS^{G12D}$;*p48Cre* mice. Further, PanINs from both $AGO2^{+/+}$;$KRAS^{G12D}$;*p48Cre* and $AGO2^{fl/fl}$;$KRAS^{G12D}$;*p48Cre* mice displayed high mucin content by Alcian blue staining[14] and similar gross weights of the pancreas, indicating indistinct phenotypes at 12 weeks (Supplementary Fig. 3a–b).

### AGO2 loss blocks PDAC progression and increases survival.
Surprisingly, mice aged over 400 days showed significantly increased pancreatic weights in both the $AGO2^{+/+}$;$KRAS^{G12D}$; *p48Cre* and $AGO2^{fl/+}$;$KRAS^{G12D}$;*p48Cre* cohort compared with $AGO2^{fl/fl}$;$KRAS^{G12D}$;*p48Cre* mice, suggestive of a higher tumor burden in mice with at least one functional allele of *AGO2* (Fig. 1d). Histology of pancreata at the 400-day time point showed early/late PanIN lesions and some PDAC development in $AGO2^{+/+}$;$KRAS^{G12D}$;*p48Cre* and $AGO2^{fl/+}$;$KRAS^{G12D}$;*p48Cre* mice with a distribution consistent with those previously reported[8,15]. However, in the $AGO2^{fl/fl}$;$KRAS^{G12D}$;*p48Cre* mice, mostly early stage PanIN lesions were observed, strikingly, with no evidence of PDAC (Fig. 1e). Occasionally, higher grade PanIN lesions were observed in $AGO2^{fl/fl}$;$KRAS^{G12D}$;*p48Cre* pancreata, but these lesions invariably showed AGO2 expression (Supplementary Fig. 4a), indicative of likely escape from Cre recombination, as has been previously noted in other contexts[8,16].

To examine the effect of *AGO2* loss on tumor-free survival, a cohort of transgenic mice was monitored over 500 days. Twelve of 12 $AGO2^{+/+}$;$KRAS^{G12D}$;*p48Cre* and 18 of 19 $AGO2^{fl/+}$; $KRAS^{G12D}$;*p48Cre* mice died over a median of 406 and 414 days, respectively, typical for a murine model expressing $KRAS^{G12D}$ in the pancreas[17,18]. Remarkably, however, all mice with homozygous *AGO2* deficiency ($AGO2^{fl/fl}$;$KRAS^{G12D}$;*p48Cre*) had survived at the 500 day cut-off (Fig. 1f). PDAC was observed in pancreata of all mice that expressed *AGO2*, but mice deficient for *AGO2* developed only early PanIN precursor lesions without progression to PDAC (Fig. 1g). Necropsies of experimental mice revealed frequent metastases and abnormal pathologies[19] in the $AGO2^{+/+}$;$KRAS^{G12D}$;*p48Cre* and $AGO2^{fl/+}$;$KRAS^{G12D}$;*p48Cre* genotypes, but $AGO2^{fl/fl}$; $KRAS^{G12D}$;*p48Cre* mice rarely showed abnormal pathologies and were without PDAC or metastases (Fig. 1g). Analyses of lungs with abnormal pathologies in two of the $AGO2^{fl/fl}$; $KRAS^{G12D}$;*p48Cre* mice (marked as gray boxes) showed a single benign lesion each, associated with AGO2 expression without indication of PDAC (Supplementary Table 2, Supplementary Fig. 4b). One mouse of the $AGO2^{fl/fl}$;$KRAS^{G12D}$;*p48Cre* genotype developed a pancreatic cyst (without AGO2 expression), histologically resembling the mucinous cystic neoplasm, and survived for 368 days (Supplementary Table 2,

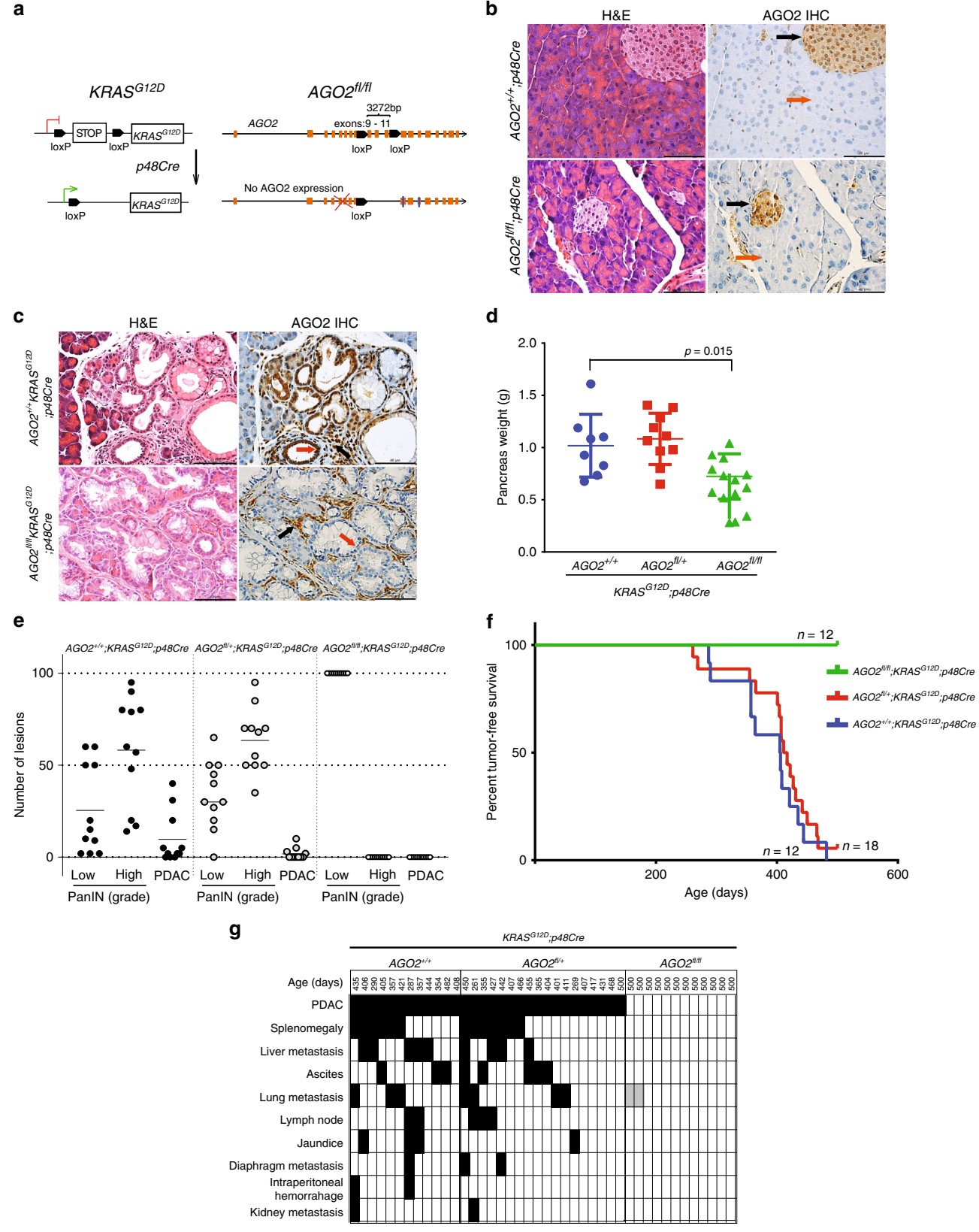

Supplementary Fig. 4b). Taken together, these data show that *AGO2* is not essential for normal pancreatic development or *KRAS[G12D]*-driven PanIN formation. Notably, however, *AGO2* is indispensable for progression of PanINs to PDAC, despite the presence of other Argonaute proteins not deleted in this model with compensatory and overlapping RNAi functions.

**PanINs with *AGO2* loss undergo oncogene-induced senescence**. Since precancerous lesions have been shown to undergo oncogene-induced senescence (OIS) in the pancreatic cancer mouse model[20], we performed OIS-associated β-galactosidase staining in pancreatic tissue sections of *AGO2[+/+];KRAS[G12D]; p48Cre* and *AGO2[fl/fl];KRAS[G12D];p48Cre* mice. As shown in

**Fig. 1 AGO2 is essential for progression of precursor PanIN lesions to PDAC. a** Schematic of the conditionally activated endogenous alleles of $KRAS^{G12D}$ and AGO2 used in the study to generate the $AGO2^{fl/fl};KRAS^{G12D};p48Cre$ experimental mice. **b** Representative images of H&E and AGO2 IHC analysis of pancreata obtained from $AGO2^{+/+};p48Cre$ and $AGO2^{fl/fl};p48Cre$ genotypes. Orange and black arrows indicate AGO2 expression in acinar cells and islets of Langerhans, respectively. Scale bar, 100 μm. **c** Representative H&E and IHC analysis for AGO2 in pancreata obtained from 12-week old mice from the $AGO2^{+/+};KRAS^{G12D};p48Cre$ and $AGO2^{fl/fl};KRAS^{G12D};p48Cre$ genotypes. Orange and black arrows indicate AGO2 staining in the PanIN and stromal regions, respectively. Scale bar, 100 μm. **d** Scatter plot showing the weight of pancreata obtained from 10 $AGO2^{+/+};KRAS^{G12D};p48Cre$, 17 $AGO2^{fl/+};KRAS^{G12D};p48Cre$, and 14 $AGO2^{fl/fl};KRAS^{G12D};p48Cre$ mice aged over 400 days. Two sided t-test was performed to determine the P value and error bars are mean values +/− SEM. **e** Histogram showing average number of early and late PanIN lesions observed in 11 mice each of $AGO2^{+/+};KRAS^{G12D};p48Cre$, $AGO2^{fl/+};KRAS^{G12D};p48Cre$, and $AGO2^{fl/fl};KRAS^{G12D};p48Cre$ genotypes at 400 days. The number of early/ late PanINs and PDAC within pancreatic sections from each animal were counted as a percentage. For $AGO2^{fl/fl};KRAS^{G12D};p48Cre$ mice, only lesions that do not express AGO2 have been included.
**f** Kaplan–Meier curve for tumor-free survival of $AGO2^{+/+};KRAS^{G12D};p48Cre$, $AGO2^{fl/+};KRAS^{G12D};p48Cre$, and $AGO2^{fl/fl};KRAS^{G12D};p48Cre$ mice aged over 500 days. **g** Chart showing PDAC (within the pancreas), the different metastatic lesions, and abnormal pathologies (black boxes) observed in each mouse of the indicated genotypes aged over 500 days. Gray boxes in the $AGO2^{fl/fl};KRAS^{G12D};p48Cre$ group indicate abnormal pathology observed at the indicated site and are addressed in further detail in Supplementary Fig. 4. The number of mice indicated in this figure represent biologically independent individuals.

Fig. 2a–b, PanINs of $AGO2^{fl/fl};KRAS^{G12D};p48Cre$ mice showed a significant increase in senescence at the early time point that dramatically increased at 500 days compared with those with AGO2 expression. Interestingly, immunoblot analysis of pancreatic tissues obtained from $AGO2^{fl/fl};KRAS^{G12D};p48Cre$ mice revealed a significant increase in phospho-ERK levels compared with $AGO2^{+/+};KRAS^{G12D};p48Cre$ mice (which progress to PDAC), indicative of hyperactive MAPK signaling downstream of RAS in the absence of AGO2 (Fig. 2c). This striking observation resembles the effects of oncogenic $BRAF^{V600E}$ in the pancreas[21]. Consistent with immunoblot analysis, phospho-ERK also showed strong and uniform IHC staining within PanINs in samples with AGO2 ablation (Supplementary Fig. 5). By contrast, phospho-ERK staining was not uniformly detected in the PDACs from $AGO2^{+/+};KRAS^{G12D};p48Cre$ mice. Thus, oncogenic KRAS-driven progression from PanIN to PDAC requires AGO2 expression to block OIS in mice. We also observed OIS with high levels of phospho-ERK staining in PanINs from human pancreatic tissue (Fig. 2d), suggesting that similar mechanisms may block PDAC development in the clinic.

Comparing immune profiles, we observed almost a fourfold increase in infiltration of $CD8^+$ T lymphocytes and 20-fold increase in natural killer (NK) cells in pancreata lacking AGO2. No significant differences in the number of $CD4^+$ Th cells or $CD68^+$ macrophages were observed between genotypes (Fig. 2e). It is interesting to note that $CD8^+$ T cells and NK cells share properties and interact to elicit potent cytotoxic activities[22]. Further, senescent PanINs lacking AGO2 showed a marked increase in the NK cell population compared with PanINs of AGO2-expressing mice (Fig. 2f–g) along the periphery and in close proximity to the PanIN lesions (Fig. 2h, Supplementary Fig. 6). Therefore, similar to other settings[23,24], the senescent phenotype in our model supports NK cell engagement.

***p53* loss bypasses the OIS associated with *AGO2* ablation.** Since p53 loss leads to evasion of senescence[25] and mutational inactivation of the *TP53* gene has been observed in approximately 75% of PDAC patients[26], we determined the role of AGO2 in the context of p53 loss. For these studies, ablation of AGO2 expression was carried out in the $KRAS^{G12D};Trp53^{fl/+};p48Cre$ (KPC) mouse model. In these mice, Cre activation simultaneously activates KRAS and reduces Trp53 levels. Tumor-free survival of chimeric mice with $AGO2^{+/+};KRAS^{G12D};Trp53^{fl/+};p48Cre$, $AGO2^{fl/+};KRAS^{G12D};Trp53^{fl/+};p48Cre$, and $AGO2^{fl/fl};KRAS^{G12D};Trp53^{fl/+};p48Cre$ genotypes was similar (Fig. 3a). PDAC and metastatic spread were also similar in all genotypes (Fig. 3b) despite efficient AGO2 ablation (Fig. 3c). Thus, a requirement for AGO2 in PDAC progression can be bypassed in a mouse model

with *TP53* aberrations[25], considered a late event in the development of pancreatic cancer[27].

**Membrane association of RAS and AGO2 in PDAC development.** Having identified an essential role for AGO2 in PDAC progression in mice, expression levels of AGO2 were next analyzed. Consistent with a role of AGO2 in KRAS-driven oncogenesis in $AGO2^{+/+};KRAS^{G12D};p48Cre$ mice, IHC analysis showed increased levels of AGO2 in PDAC and metastatic tissues as compared with early PanIN lesions (Fig. 4a). Transcript analysis in human pancreatic cancer suggested a significant increase in AGO2 expression in PDAC compared with normal pancreas (Supplementary Fig. 7a). To extend these observations at the protein level, we performed a systematic IHC analysis of a human pancreatic tissue microarray (TMA), comprising 44 duplicate pancreatic tissue cores, including PanIN, PDAC, and metastatic PDAC samples. AGO2 expression was remarkably higher in PDAC and metastatic PDAC cells compared with PanINs (Fig. 4b), and this increase was statistically significant (Fig. 4c). These data show that AGO2 protein levels are elevated with disease progression and suggest an important role for AGO2 in pancreatic cancer development in humans.

Considering that RAS is known to localize to the plasma membrane[2,28], we tested if RAS and the RAS-AGO2 interaction could be localized at the plasma membrane in the mouse models and human tissues. Since most commercial KRAS-specific antibodies have been shown to be unsuitable for IHC or immunofluorescence (IF)[29], we tested RAS10, a pan-RAS monoclonal antibody, and observed specific staining only in RAS expressing cells (Fig. 5a, Supplementary Fig. 7b–c). Surprisingly, relative to the surrounding normal tissue, IHC and IF analysis of mouse pancreatic tissues with this antibody detected high membranous RAS expression within the PanINs (Fig. 5b). Pre-incubation of the antibody with RAS peptides spanning the antibody epitope abrogated RAS staining (Fig. 5c). To corroborate the finding that RAS IHC and IF staining were primarily restricted to oncogenic KRAS-driven PanINs, we performed RNA in situ hybridization (RNA-ISH) using KRAS-targeted RNA probes (Supplementary Fig. 7d). As shown in Supplementary Fig. 7e, we observed KRAS transcripts restricted to the ducts of pancreatic lesions. The elevated KRAS transcript expression in PanINs is consistent with a recent study reporting increased oncogenic KRAS transcripts in engineered mouse models[30]. We also validated the AGO2 monoclonal antibody using pancreatic tissue from experimental mice (Fig. 5d, Supplementary Fig. 2).

Next, we assessed whether RAS and AGO2 co-localized in pancreatic tissues during PDAC progression. IF staining of normal acinar cells in the mouse pancreas displayed low and diffuse cytoplasmic staining of RAS and minimal expression of

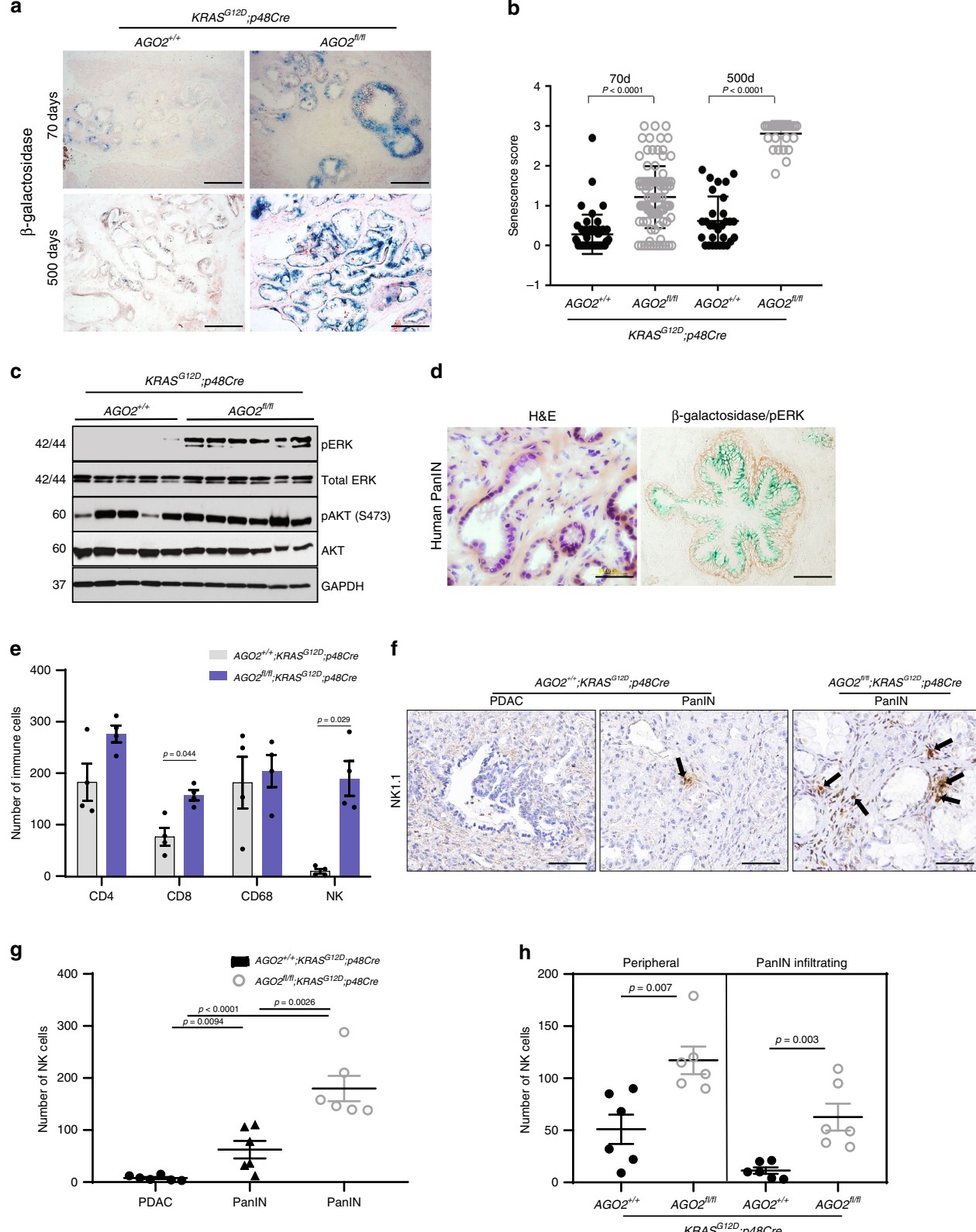

AGO2, with a low measure of co-staining pattern (Pearson's correlation for co-localization, PCC = 0.1) (Fig. 5e). Interestingly, as shown in Fig. 5e, RAS expression increased in PanINs, which further increased in PDAC (Supplementary Fig. 7f). In a parallel manner, AGO2 expression progressively increased in PanIN and PDAC tissues (Supplementary Fig. 7g), with a concomitant increase in plasma membrane localization and co-staining

patterns with RAS (PanIN PCC = 0.5; PDAC PCC = 0.7). RAS staining could also be detected in the PanIN lesions of *AGO2^{fl/fl}; Kras^{G12D};p48Cre* mice (Supplementary Fig. 8a). Localization of AGO2 to the plasma membrane was independently confirmed by analyzing co-localization with the membrane marker, E-cadherin (PCC = 0.43; Supplementary Fig. 8b). Furthermore, in pancreatic tissue obtained from normal mice treated with caerulein to

**Fig. 2 *AGO2* loss prevents PanIN to PDAC progression through OIS. a** β-galactosidase staining of pancreatic sections from *AGO2*[+/+];*KRAS*[G12D];*p48Cre* and *AGO2*[fl/fl];*KRAS*[G12D];*p48Cre* mice at 70- and 500-day time points. Scale bar, 100 μm. **b** Scatter plot showing β-galactosidase staining in low grade PanINs. Data are from 47 PanINs from *AGO2*[+/+];*KRAS*[G12D];*p48Cre* and 98 PanINs from *AGO2*[fl/fl];*KRAS*[G12D];*p48Cre* from four individual mice at the 70-day time point and 30 PanINs from three individual mice at 500-day time points. Intensity of staining and percent cells within 30 low grade PanINs were used to determine the senescence score = intensity × percent positive cells. *p* values were determined using a two sided *t*-test. Data are presented as mean values +/− SEM. **c** Immunoblot analysis of RAS-driven MAPK (indicated by pERK) and PI3K (indicated by pAKT) signaling from individual pancreata obtained from mice of the indicated genotypes, aged to 400 days. Numbers on the left indicate protein molecular weights in kDa. **d** Representative images of H&E staining (left) and dual staining for β-galactosidase and phospho-ERK (right) in human pancreatic tissue with PanINs (representative staining of at least 10 PanINs from two patients). Scale bar, 40 μm. **e** Immune profile of lesions from the indicated genotypes. Ten consecutive fields (20x magnification) from four individual mice were assessed for the indicated IHC markers that distinguish immune cell populations. Significant *p* values are indicated and were determined using two tailed *t*-test. **f** Representative images of NK1.1-positive NK cells surrounding PanIN and PDAC lesions within the indicated genotypes. Scale bar, 50 μm. **g** Plot showing NK cell number in PanIN/PDAC lesions within the indicated genotypes. Pancreatic tissues from six mice were analyzed for NK1.1 IHC-positive NK cells. Two tailed *t*-test was used to determine *p* values. **h** Scatter plot showing peripheral and PanIN infiltrating NK cell count from PanINs in the indicated genotypes. Counts were obtained from 10 consecutive fields from six mice at ×20 magnification, and the indicated *p* values were determined using two tailed *t*-test. In relevant panels, data are presented as mean values +/− SEM.

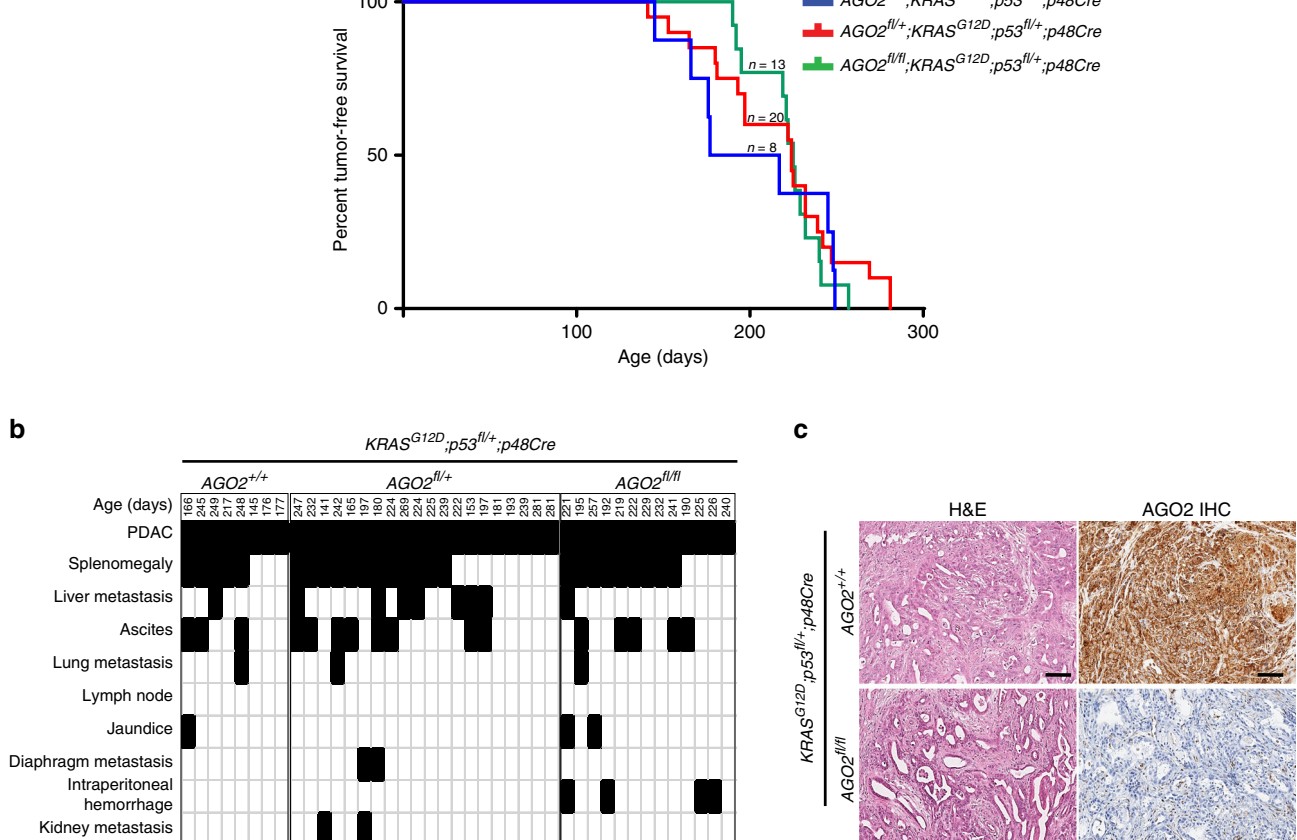

**Fig. 3 p53 loss bypasses requirement for *AGO2* during PDAC progression. a** Kaplan–Meier tumor-free survival of *AGO2*[+/+];*KRAS*[G12D];*Trp53*[fl/+];*p48Cre*, *AGO2*[fl/+];*KRAS*[G12D];*Trp53*[fl/+];*p48Cre*, and *AGO2*[fl/fl];*KRAS*[G12D];*Trp53*[fl/+];*p48Cre* mice. **b** Chart showing PDAC (within the pancreas), the different metastatic lesions, and abnormal pathologies (black boxes) observed in each mouse of the indicated genotypes. **c** Representative H&E and AGO2 IHC in the indicated genotype. Scale bar, 100 μm.

induce pancreatitis[31], we observed wild-type RAS localization at the membrane without AGO2 co-staining (Supplementary Fig. 8c). This suggests specificity of the RAS-AGO2 co-staining observed during oncogenic KRAS-driven PDAC development. Importantly, extending the IF analysis to human pancreatic tissues, we observed a similar pattern of localization of RAS and AGO2 with increased RAS-AGO2 co-staining signals at the plasma membrane associated with pancreatic cancer progression

(PCC, normal to PDAC increased from 0.1 to 0.5, respectively) (Fig. 5f).

For a direct assessment of the RAS-AGO2 interaction at single molecule resolution, we performed proximity ligation assays (PLA)[32,33] using the RAS and AGO2 antibodies validated earlier (Fig. 5a, Supplementary Figs. 2, 8d). As shown in Fig. 5g, PLA signals indicative of RAS-AGO2 interaction were observed at the plasma membrane within PanINs arising in *AGO2*[+/+];*KRAS*[G12D];*p48Cre*

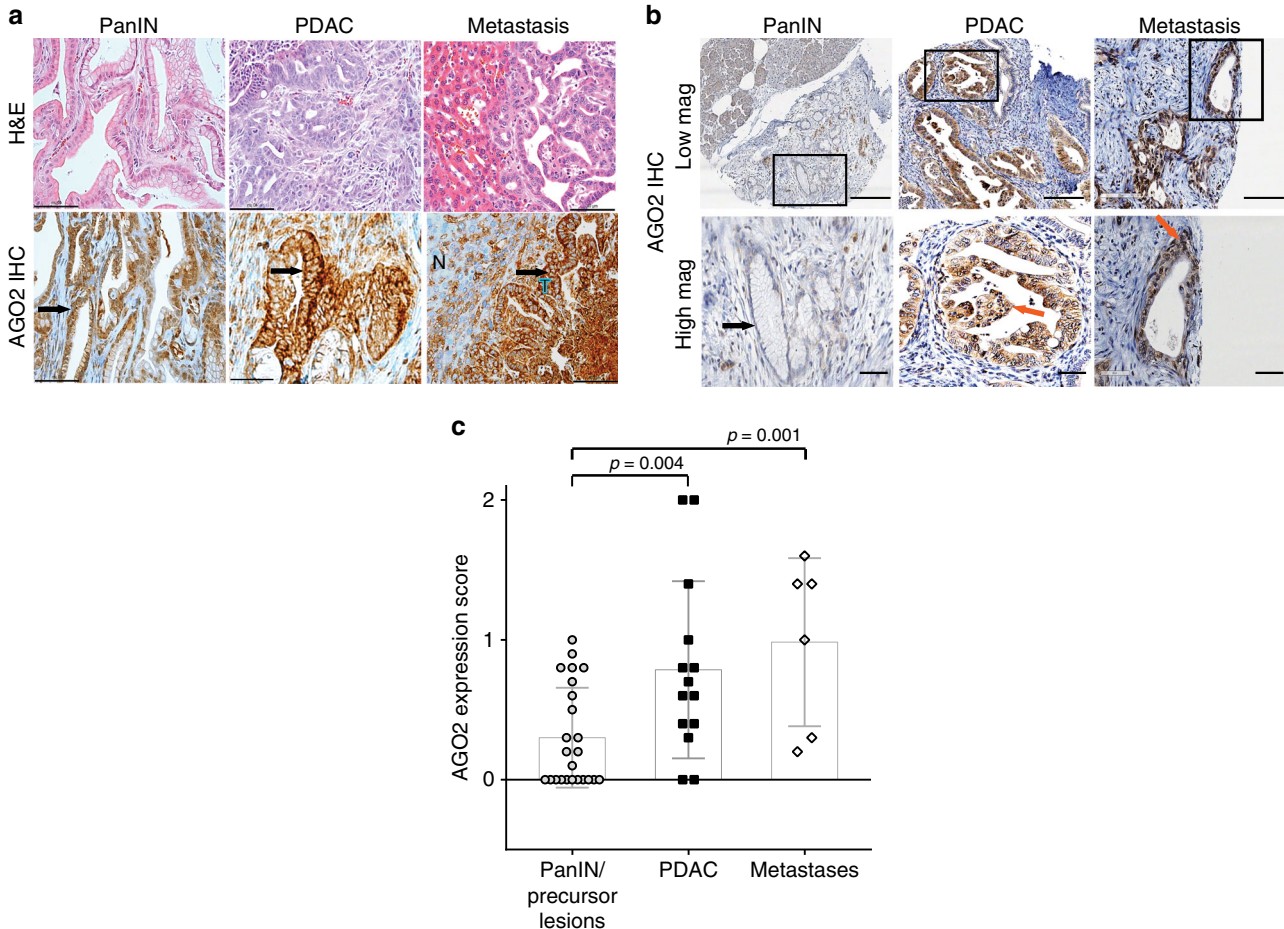

**Fig. 4 Increased AGO2 expression during mouse and human PDAC progression. a** Representative images of AGO2 IHC analysis within an individual $AGO2^{+/+};KRAS^{G12D};p48Cre$ mouse showing increased AGO2 expression in PDAC and metastasis compared with PanIN lesions. Arrows point to PanIN, PDAC, or metastatic PDAC in respective panels. In the metastasis panel, N = normal liver and T = tumor. Scale bar, 40 μm. **b** Representative images of IHC analysis for AGO2 expression in human PDAC progression showing elevated AGO2 protein expression in PDAC and metastatic tissue. Lower panels show higher magnifications of areas marked in the upper panels. Scale bars in the top and bottom panels are 200 and 80 μm, respectively. Arrows point to PanIN and PDAC. **c** Box and scatter plot showing AGO2 expression on a human tissue microarray (TMA) containing 44 human pancreatic tissue samples (24 precancerous, 14 PDAC, and six metastatic PDAC lesions), as determined by IHC analysis. Each sample was scored for intensity of stain and percent tumor cells staining for AGO2, and the final score = intensity × percent positive cells. $p$ values were determined using a two sided $t$-test. Data are presented as mean values $+/-$SEM.

but not in $AGO2^{fl/fl};KRAS^{G12D};p48Cre$ mice. This further corroborates the IF analyses and provides evidence of membranous RAS-AGO2 interaction. Together, these data indicate that during pancreatic cancer development, AGO2 localizes at the plasma membrane, the site of RAS activity[2,28], and substantiates a role for $AGO2$ in the progression of PanINs to PDAC. Interestingly, in the KPC model described above (Fig. 3c, Supplementary Fig. 9a), AGO2 expression also increased during PDAC progression in AGO2-sufficient pancreata (Supplementary Fig. 9b). Significant overlapping membranous signals for RAS and AGO2 (PCC = 0.7) were observed in PDAC lesions from $AGO2^{+/+};KRAS^{G12D};Trp53^{fl/+};p48Cre$ mice (Supplementary Fig. 9c).

**AGO2 regulates expression of microRNAs that control OIS.** Considering a central role for AGO2 in the RNAi pathway, we compared the microRNA expression profiles of pancreata from $AGO2^{+/+};KRAS^{G12D};p48Cre$ and $AGO2^{fl/fl};KRAS^{G12D};p48Cre$ mice at the 500-day time point. Among the small number of microRNAs that showed differential expression between the two genotypes, the miR-29 and miR-30 families of microRNAs were

significantly downregulated in pancreata expressing $AGO2$ (Fig. 6a, Supplementary Fig. 10a). Expression of this family of microRNAs was relatively upregulated within the PanINs lacking $AGO2$. These Rb-regulated microRNAs have been strongly associated with senescence[34], suggesting that their expression in the setting of $AGO2$ loss contributes to the OIS phenotype. A similar pattern of regulation was observed for members of the tumor suppressive $let$-7 family of microRNAs known to regulate cell proliferation[35] and differentiation[36]. On the other hand, oncogenic miR-21 was expressed to the same extent in the two genotypes. However, in the KPC model, downregulation of the miR-29, miR-30, and $let$-7 microRNA families was independent of $AGO2$ status (Supplementary Fig. 10b), allowing for progression to PDAC even in the absence of $AGO2$.

Transcriptomic profiles of pancreata obtained from the two genotypes showed that the block in PDAC progression in $AGO2$ loss was associated with reduced gene set enrichment scores for E2F targets, G2/M checkpoint controls, and canonical oncogenic KRAS signaling (Fig. 6b, Supplementary Fig. 11). Surprisingly, these transcriptional changes were accompanied with an increase in oxidative phosphorylation, considered conducive for PDAC

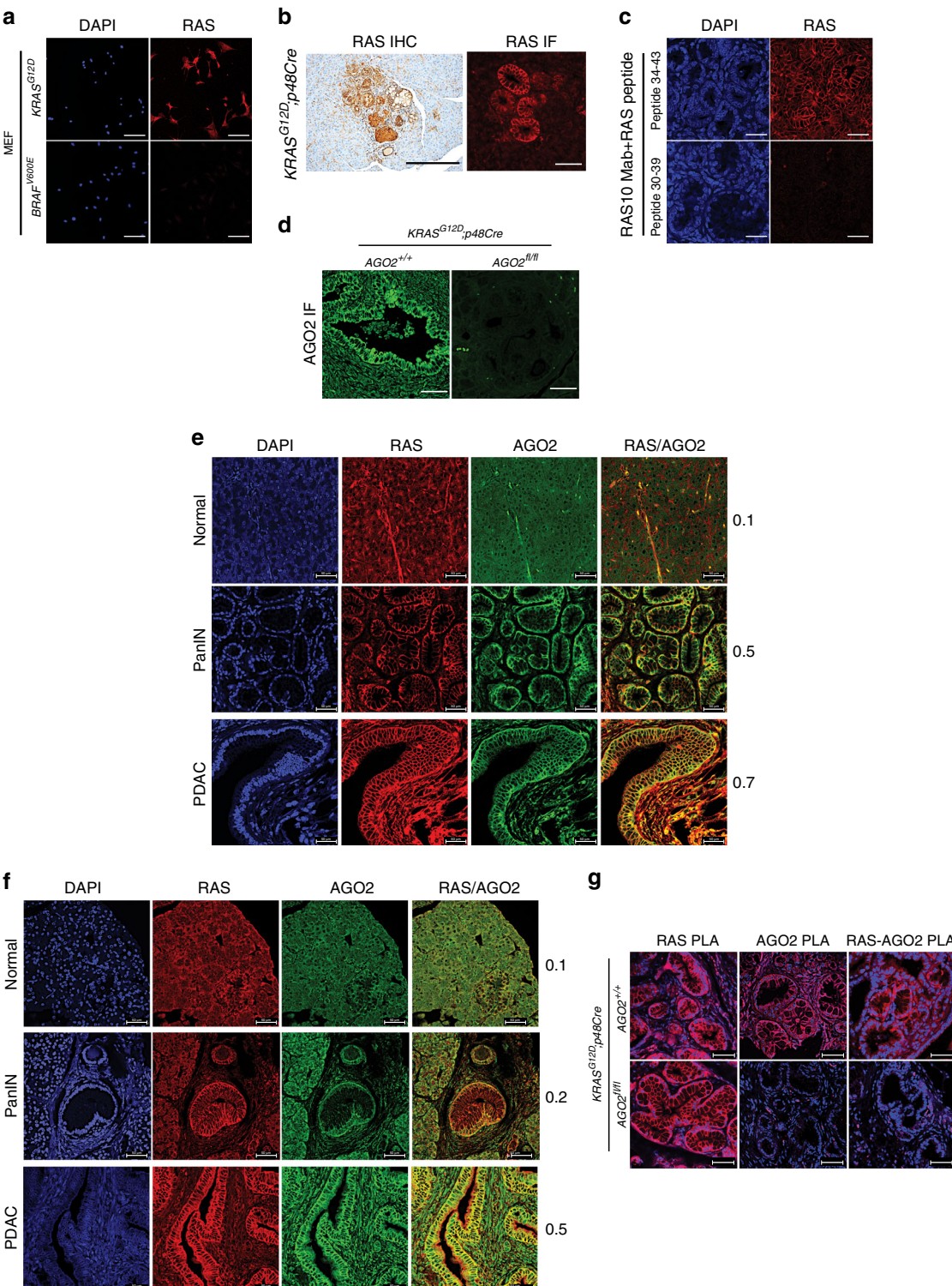

development[37]. These data suggest a temporal requirement for *AGO2* for the biogenesis of select microRNAs to control PDAC progression.

**The KRAS-AGO2 interaction limits RAS activation.** Next, we sought to explore how AGO2 alters the EGFR-RAS signaling axis for two reasons: (1) EGFR has been shown to be essential for PanIN formation in the *KRAS^{G12D}*-driven pancreatic mouse model[7,8,38], and (2) EGFR activation has been shown to directly

inhibit AGO2 function through phosphorylation of its tyrosine 393 residue[12]. Immunoblot analysis of pancreatic tissues from 12-week-old mice with *AGO2^{+/+};KRAS^{G12D};p48Cre*, *AGO2^{fl/+}; KRAS^{G12D};p48Cre*, and *AGO2^{fl/fl};KRAS^{G12D};p48Cre* genotypes showed a marked increase in AGO2 levels relative to normal pancreata (Supplementary Fig. 12a) concordant with IF analysis (Fig. 5e). Consistent with published studies[7,8], total EGFR levels were also elevated in *KRAS^{G12D}* mice irrespective of the *AGO2* genotype (Supplementary Fig. 12a). However, in early PanINs initiated by oncogenic KRAS, significantly higher levels of

**Fig. 5 Increased membrane co-localization of RAS and AGO2 during PDAC progression. a** RAS10 (panRAS) antibody specificity for IHC and IF analyses was determined by staining RASless MEFs rescued by either oncogenic *KRAS* or *BRAF^V600E*. Scale bar, 100 μm. **b** Membranous RAS staining in 10-week-old PanINs of mouse tissues expressing oncogenic *KRAS* using either IHC (left) or IF (right). Scale bars, 50 μm. **c** Peptide competition assay to demonstrate specificity of the RAS10 antibody in mouse tissues expressing oncogenic *KRAS*. Representative IF images using the RAS10 antibody pre-incubated with RAS peptide spanning the antibody epitope 30-39aa and control overlapping RAS peptide spanning 34-43aa. Scale bar, 50 μm. **d** Representative images of AGO2 IF analysis in pancreatic tissues from *AGO2^+/+;KRAS^G12D;p48Cre* and *AGO2^fl/fl;KRAS^G12D;p48Cre* mice. Scale bar, 50 μm. **e** Representative images of IF analysis for RAS and AGO2 through PDAC progression in the *AGO2^+/+;KRAS^G12D;p48Cre* mice. Scale bar, 50 μm. **f** Representative images of IF analysis of human pancreatic tissue on a TMA showing co-localization of AGO2 and RAS in PanIN and PDAC cells. For (**e**) and (**f**), numbers adjacent to merged images indicate the Pearson's coefficient of co-localization (PCC) of RAS-AGO2 signals at the membranous regions (where 0 is no overlap and 1 is complete overlap). PCC was determined using co-localization signals of at least 50 cells in three distinct areas representative of normal acinar, PanIN, PDAC, or metastases. Scale bar, 50 μm. **g** Representative images of Proximity Ligation Assay (PLA), performed to detect either RAS (RAS PLA) or AGO2 (AGO2 PLA) expression and the RAS-AGO2 interaction (RAS-AGO2 PLA) within PanIN lesions of *AGO2^+/+;KRAS^G12D;p48Cre* (upper panel) and *AGO2^fl/fl; KRAS^G12D;p48Cre* (lower panel) mice. PLA signals appear as red dots around DAPI stained nuclei in blue. Scale bar, 50 μm.

phospho-EGFR (Y1068) were observed in pancreatic tissues of *AGO2^fl/fl;KRAS^G12D;p48Cre* mice (Fig. 6c, Supplementary Fig. 12a-b), indicating activated EGFR signaling in the absence of *AGO2* expression. IHC analysis confirmed the elevated phospho-EGFR levels observed in tissue lysates were restricted to PanIN lesions of *AGO2^fl/fl;KRAS^G12D;p48Cre* mice (Fig. 6d). IHC of total EGFR showed no significant difference in expression in pancreatic tissues between genotypes (Fig. 6d, Supplementary Fig. 13a–b). As previously noted, irrespective of *AGO2* genotype, lesions from later time points showed a marked reduction in total EGFR levels[7,8] in mice (and with disease progression in human tissue), further supporting the significance of EGFR signaling in the early stages of disease (Supplementary Fig. 13a,c). Importantly, immunoblot analysis showed that EGFR activation was accompanied with a remarkable increase in total RAS levels but not oncogenic KRAS^G12D levels (Fig. 6c, Supplementary Fig. 12), raising an intriguing possibility that signaling in early stage PanINs is along the EGFR-wild-type RAS axis.

To investigate this further, we isolated pancreatic ducts from 12-week-old *AGO2^+/+;KRAS^G12D;p48Cre* and *AGO2^fl/fl; KRAS^G12D;p48Cre* mice and cultured them as organoids[39] in the absence of EGF (Supplementary Fig. 14). Immunoblot analysis showed increased levels of phospho-EGFR and total RAS in the organoids with *AGO2* loss, while KRAS^G12D expression showed no change (Fig. 6e), mirroring the observations from pancreatic tissue lysates. Given that AGO2 is a direct phosphorylation substrate of the EGFR kinase[12], our experiments define a reverse feedback upregulation of phospho-EGFR via AGO2[40–43].

To estimate the levels of activated wild-type RAS and oncogenic KRAS due to EGFR activation in *AGO2* loss, we performed the RAF-binding domain (RBD) assay using isoform specific antibodies. As shown in Fig. 6e, significant increases in activated RAS were readily detected using pan-RAS and KRAS-specific antibodies. A modest increase in KRAS^G12D-GTP levels was also observed. These observations reveal that *AGO2* ablation activates EGFR signaling and results in increased GTP loading of both the wild-type and oncogenic KRAS.

Next, we monitored the extent of wild-type KRAS-GTP and KRAS^G12D-GTP levels upon treatment of the organoids with erlotinib. Remarkably, erlotinib treatment of organoids expressing *AGO2* showed an increase in KRAS^G12D-GTP levels. On the other hand, EGFR inhibition in *AGO2*-deficient organoids had no effect on the increased KRAS^G12D-GTP levels (Fig. 6f). Conversely, in the absence of *AGO2*, EGFR inhibition dramatically reduced wild-type KRAS-GTP levels. The total KRAS-GTP levels in *AGO2*-sufficient organoids tracked with those of KRAS^G12D-GTP levels upon erlotinib treatment. These experiments clarify the role of AGO2 as a regulator of KRAS activity. In its absence, both oncogenic and wild-type RAS forms are activated, yet unlike

oncogenic KRAS, activated wild-type KRAS is sensitive to EGFR inhibition. Further, the activation of EGFR, wild-type KRAS, and downstream ERK, all remained sensitive to erlotinib treatment (Fig. 6f).

To probe if *AGO2* loss activates wild-type RAS even in the absence of mutant KRAS, we performed immunoblot analysis and RAS activation assays using *AGO2^−/−* MEFs that do not harbor any form of oncogenic RAS[44]. As shown in Fig. 6g, *AGO2^−/−* MEFs also exhibit increased phospho-EGFR and wild-type RAS levels along with elevated wild-type RAS-GTP levels, which were significantly reduced when rescued with *AGO2*. We then made stable lines expressing either vector, full-length wild-type AGO2, or RAS binding-deficient AGO2 (K112A/E114A) to interrogate the specific effects of the KRAS-AGO2 interaction on RAS signaling. As shown in Fig. 6h, both wild-type AGO2 and AGO2^K112A/E114A decreased phospho-EGFR activation. While the wild-type form of AGO2 limited RAS activation, the AGO2 mutant lacking RAS-binding residues sustained RAS activation. Further, increased pERK activation observed in the *AGO2^−/−* MEFs could be rescued with AGO2 but not the RAS binding-deficient mutant (Fig. 6h). Using diverse models, these data define AGO2 as a key regulator that limits KRAS activation and downstream effector engagement through its direct interaction with KRAS.

To further explore how AGO2 could regulate KRAS-GTP levels, we performed biochemical assays that monitor the levels of KRAS-GTP in the presence of known RAS regulators. Specifically, using purified catalytic domains of Neurofibromin 1-GAP and SOS1-GEF, we measured KRAS-GTP levels in the presence or absence of AGO2. As shown in Fig. 6i, AGO2 had no effect on the intrinsic GTPase activity of KRAS and did not alter NF1-GAP activity on KRAS. However, SOS1-mediated nucleotide exchange on wild-type RAS was significantly reduced in the presence of AGO2 (Fig. 6j) in a dose-dependent manner (Fig. 6k). In similar assays, oncogenic forms of KRAS were resistant to both GAP and GEF activity. Since both SOS and AGO2 compete for binding to the KRAS Switch II domain through Y64, our data predicts that the KRAS-AGO2 interaction limits wild-type RAS activation by competitively blocking SOS association (Fig. 6). Thus, in addition to the microRNA and transcriptomic changes observed with *AGO2* ablation, loss of the KRAS-AGO2 interaction likely leads to increased ERK activation associated with the OIS phenotype.

**AGO2^Y393 phosphorylation disrupts the KRAS-AGO2 interaction.** Considering that *AGO2* loss leads to increased signaling via EGFR and wild-type RAS through increased access of SOS to KRAS (Fig. 6j–k), we posited that AGO2 binding to KRAS may also represent a rate limiting step in the activation of wild-type KRAS during growth factor stimulation. To explore this premise, we assayed for KRAS-AGO2 interaction across a panel of cell

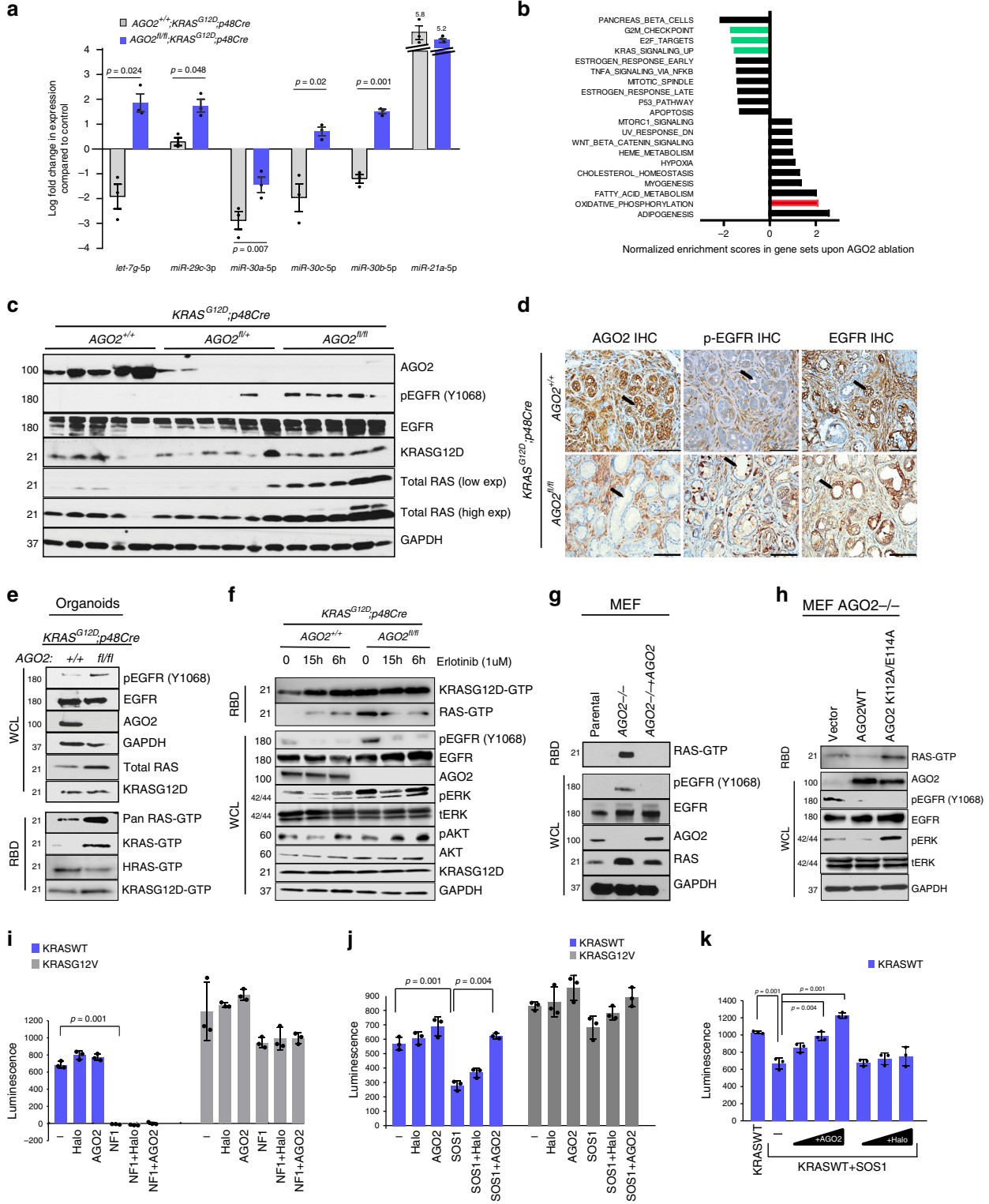

lines expressing wild-type or mutant *RAS* stimulated with EGF. Interestingly, EGF stimulation resulted in a dramatic decline in KRAS-AGO2 interaction in cells with wild-type *KRAS*, as observed in MCF-7, PC3, A375, and HeLa cells (Fig. 7a, Supplementary Fig. 15a–b). In contrast, EGF stimulation of cells harboring oncogenic *KRAS*, including A549 (*KRAS^{G12S}*), MIA PaCa-2 (*KRAS^{G12C}*), and Capan-1 (*KRAS^{G12V}*), retained binding of endogenous KRAS and AGO2 despite activation of the EGFR/

MAPK/AKT pathway (Fig. 7b, Supplementary Fig. 15c). Disruption of the wild-type RAS-AGO2 interaction was also observed when HEK293 (wild-type *KRAS*) cells expressing FLAG-tagged AGO2 were stimulated with EGF; the interaction was rescued by treatment of cells with erlotinib (Fig. 7c). This strongly suggests that EGFR kinase activity was critical for the disruption of the wild-type KRAS-AGO2 interaction. In contrast, DLD-1 cells harboring mutant *KRAS^{G13D}* showed no loss of

**Fig. 6 AGO2 modulates microRNAs and limits RAS activation to control OIS. a** Plot showing relative expression of microRNAs from pancreata obtained from three mice from each of the indicated genotypes at the 500-day time point. Log fold change values were generated relative to microRNA expression in three *p48Cre* mice (used as reference). Two sided *t*-tests were performed to determine the *P* value and error bars are mean values +/− SEM. **b** GSEA (Gene Set Enrichment Analysis) of transcriptional changes significantly enriched (FDR value < 0.05) in pancreatic tissue. Green and red bars represent relevant gene sets discussed in the main text. **c** Immunoblot analysis from individual pancreata obtained from 12-week-old mice of the indicated genotypes. **d** Representative images of IHC analysis in PanINs of 12-week-old mice in the indicated genotypes. Arrows indicate PanINs. Scale bar, 100 μm. **e** Immunoblot analysis of pancreatic ductal organoids obtained from 12-week-old *AGO2⁺ᐟ⁺;KRAS^{G12D};p48Cre* and *AGO2^{fl/fl};KRAS^{G12D};p48Cre* mice. Total RAS-GTP was determined using the RAF binding assay (RBD) followed by immunoblotting with indicated antibodies. **f** Immunoblot analysis of pancreatic organoids upon treatment with erlotinib at 6 h and 15 h time points. **g** Immunoblot analysis of parental, *AGO2−/−*, and *AGO2−/− + AGO2* mouse embryonic fibroblasts (MEF). RAS-GTP levels were determined by the RAF binding assay. **h** Immunoblot analysis of *AGO2−/−* MEFs stably expressing vector, wild-type AGO2, and AGO2^{K112A/E114A}. **i** Full-length wild-type KRAS and KRAS^{G12V} proteins were incubated with NF1-GTPase activating protein (GAP) or **j** SOS1 (guanine exchange factor) in the presence or absence of AGO2, and the levels of free GTP were analyzed (as a luminescence-based readout for GTP hydrolysis). Halo protein was used as a control. **k** Wild-type KRAS was incubated with SOS1 in the presence of increasing concentrations of either AGO2 or Halo protein prior to measurement of free GTP levels. Significance was assessed in (**i**–**k**) using Welch's two tailed test to determine *p* values. In relevant panels of this figure, data are presented as mean values +/− SEM. Numbers on the left of the immunoblots in this figure indicate protein molecular weights in kDa.

KRAS and AGO2 association either by EGF or erlotinib treatment (Fig. 7d).

To test if the previously identified site of EGFR-mediated phosphorylation[12] on AGO2 at tyrosine 393 has a role in binding to KRAS, we analyzed the ability of a phosphorylation-deficient AGO2^{Y393F} mutant to bind RAS under different conditions. In HEK293 (wild-type *KRAS*) cells, EGF stimulation led to dissociation of wild-type AGO2 from RAS, but the AGO2^{Y393F} mutant continued to bind RAS with or without EGFR activation (Fig. 7e), indicating that phosphorylation of this residue is critical for dissociation. Expression of these *AGO2* constructs in MIA PaCa-2 (*KRAS^{G12C}*) cells showed no discernible change in RAS binding upon EGFR activation (Fig. 7f).

Next, we treated oncogenic KRAS-expressing cells with $H_2O_2$, known to inactivate tyrosine phosphatases by oxidation and, thus, activate EGFR[45]. In both H358 and Mia PaCa-2 cells expressing oncogenic KRAS, the KRAS-AGO2 interaction could be readily disrupted following $H_2O_2$ treatment (Fig. 7g–h). This disruption was also found to be dependent on the Y393 phosphorylation site of AGO2, since mutant AGO2^{Y393F} remained recalcitrant to $H_2O_2$ treatment in Mia PaCa-2 cells (Fig. 7i).

To track the localization of the RAS-AGO2 interaction upon growth factor activation, we performed PLA on cells expressing wild-type or mutant *KRAS*. The use of either RAS or AGO2 antibodies alone did not show signals for the RAS-AGO2 PLA (Supplementary Fig. 16a). Interestingly, serum-starved PC3 cells showed increased membrane localization of both RAS and AGO2 proteins contributing to the increase in membrane-localized RAS-AGO2 PLA signals under these conditions (Fig. 7j, upper panels). RAS-AGO2 interaction PLA signals were significantly reduced upon EGF stimulation and restored to levels observed under serum-sufficient conditions. IF analyses also showed a similar pattern of RAS-AGO2 co-staining under these different culture conditions (Supplementary Fig. 16b). A similar pattern of RAS-AGO2 interaction PLA signals was observed in wild-type *RAS* expressing MCF-7 cells (Fig. 7j, panel I). In contrast, both HCT116 and H358 cells (Fig. 7j, panels II and III), expressing oncogenic forms of *KRAS*, showed higher basal levels of RAS-AGO2 PLA signals compared with wild-type *RAS* expressing cells that remained consistent under different cell culture conditions. Combined, these data suggest that the wild-type KRAS-AGO2 interaction at the membrane is sensitive to EGF-stimulated phosphorylation of AGO2^{Y393}, while the oncogenic KRAS-AGO2 interaction is unaffected by ligand-activated EGFR. These data suggest that the Y393 phosphorylation site in AGO2 determines binding of both wild-type and oncogenic forms of KRAS; however, while AGO2 phosphorylation in wild-type RAS

expressing cells can be achieved by EGF stimulation, mutant RAS-expressing cells require sustained EGFR activation through inhibition of tyrosine phosphatases.

**ARS-1620 disrupts the oncogenic KRAS-AGO2 interaction.** Finally, we tested if direct targeting of oncogenic KRAS could affect the endogenous mutant KRAS-AGO2 interaction. Interestingly, the mutant KRAS-AGO2 interaction was disrupted when H358 (Fig. 8a) and MIA-PaCa-2 cells (Fig. 8b), harboring *KRAS^{G12C}* mutant alleles, were treated with ARS-1620[46], a covalent G12C inhibitor. The disruption of endogenous KRAS^{G12C}-AGO2 interaction in these cells was concentration-dependent and reflects the differential sensitivities of the two cell lines to ARS-1620[47]. In a similar assay, ARS-1620 treatment had no effect on the KRAS^{G12D}-AGO2 interaction in Panc 05.04 (Fig. 8c) or Panc 10.05 cells (Fig. 8d). Given that ARS-1620 binds an allosteric Switch II pocket (SW-IIP)[47] on GDP-loaded KRAS^{G12C}, the disruption of KRAS^{G12C}-AGO2 binding provides orthogonal evidence that AGO2 makes contact with the Switch II region in KRAS. This data also proves that besides SOS, the easily detectable, endogenous membrane-bound KRAS^{G12C}-AGO2 interaction is an additional target of G12C inhibitors.

## Discussion

GEMMs mirror the stepwise progression of human pancreatic cancer, starting with benign precursor lesions (PanINs) driven by mutant *KRAS*[18,48,49]. Here, a GEMM of *AGO2* loss delineates pancreatic cancer development into two distinct phases (Fig. 9). The first phase of PanIN development is triggered by oncogenic *KRAS* and depends on EGFR-RAS mediated proliferation; this is followed by a second phase of PDAC progression that requires AGO2 expression to overcome OIS. While low-grade PanINs have been known to undergo senescence[20], our model represents the first instance where loss of a direct interactor of KRAS induces OIS to abrogate PDAC progression. Despite the increased presence of NK cells[50], the senescent cells are not cleared, allowing increased lifespan of mice lacking *AGO2*. Interestingly, the OIS phenotype is characterized by both the limited RNAi activity of AGO2 (to downregulate miRNAs that control cell proliferation/senescence, such as the *let-7* and miR-29/30 families[51–54]) and a concomitant increase in oncogenic and wild-type KRAS activity and downstream effector signaling. While the microRNA regulation may be a result of the complete lack of *AGO2* expression, the effects on RAS signaling can be attributed to the KRAS-AGO2 interaction. Notably, other members of the Argonaute family fail to compensate for these functions,

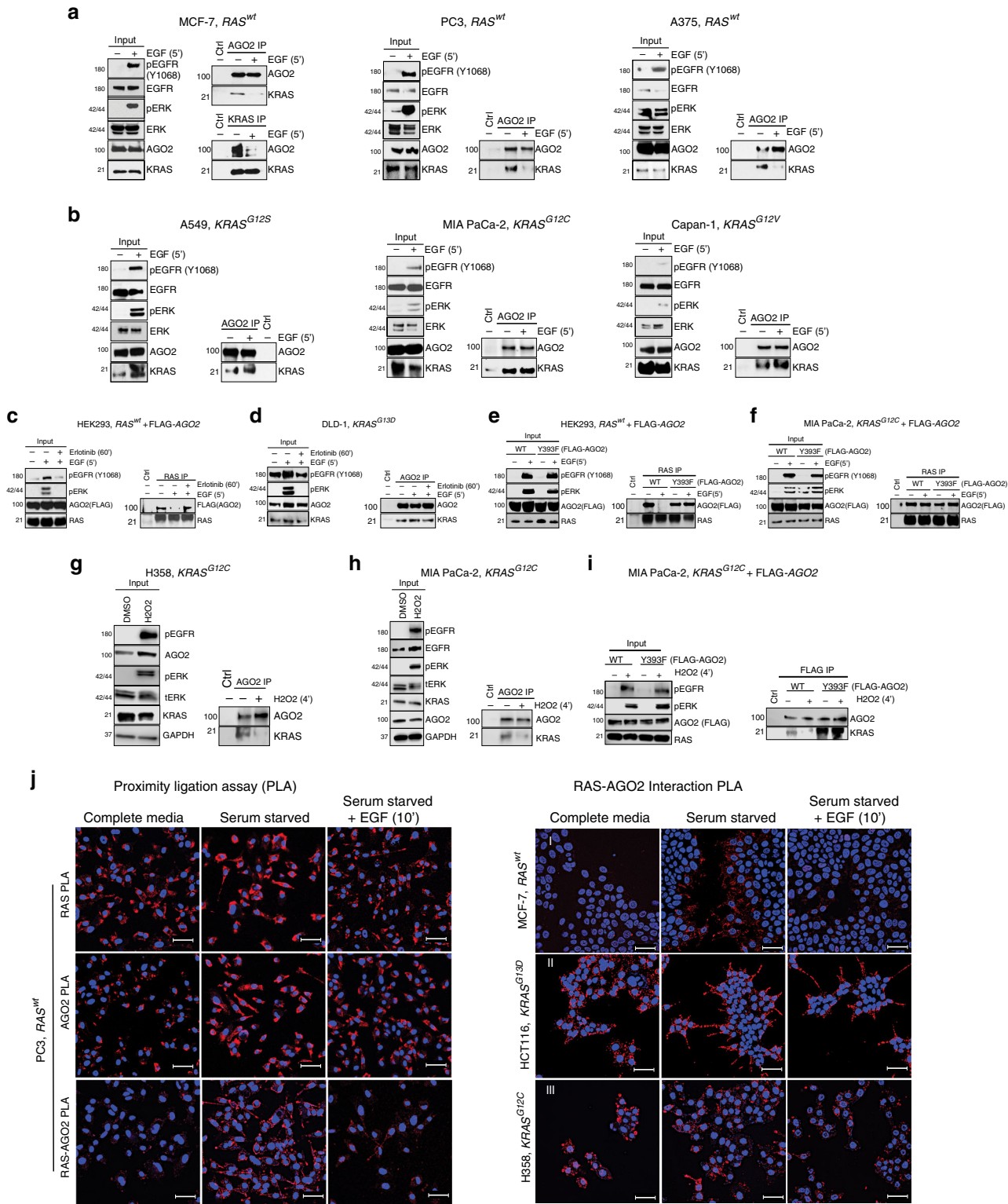

reiterating the specific role of AGO2 in KRAS-driven pancreatic cancer. Since the microRNAs identified here also regulate pancreatic autophagic process[55] and OIS[56], it will be interesting to determine the role of AGO2 in autophagy.

Through these studies, we show that AGO2 is localized to the plasma membrane, a known site for KRAS activity but not RNA silencing activity[57–60]. Importantly, we show that AGO2 expression at the membrane increased during PDAC progression, both in mouse and human PDAC, highlighting membrane

redistribution of AGO2 in clinical disease progression. Given that posttranscriptional regulation can involve dynamic polarization of mRNA targets[61], it will be interesting to understand the effects of AGO2 membrane localization on its RNAi function to control translation. Functionally, we find that p53 loss precludes a requirement for both *AGO2* and its RNAi activity, yet, like in the human tissues, RAS/AGO2 membrane localization is retained in this model. Since aberrations in *KRAS* and *p53* do not occur simultaneously, the mouse model with p53 loss represents a

**Fig. 7 Phosphorylation of AGO2$^{Y393}$ disrupts its interaction with KRAS. a** Immunoprecipitation (IP) of endogenous AGO2 upon EGF stimulation (5') in the indicated cancer cells expressing wild-type *RAS* followed by immunoblot analysis of KRAS. For MCF7 cells, endogenous co-IP analysis was performed using both AGO2 and KRAS-specific antibodies. For each cell line and panel in this figure, MAPK activation and levels of various proteins are shown as input blots. **b** IP of endogenous AGO2 upon EGF stimulation (5'), in the indicated cancer cells harboring different *KRAS* mutations, followed by immunoblot analysis of KRAS. **c** Co-IP and immunoblot analysis of RAS and AGO2 upon EGF stimulation of HEK293 (wild-type *KRAS*) cells expressing FLAG-AGO2 or (**d**) DLD-1 (*KRAS$^{G13D}$*) cells in the presence or absence of erlotinib. **e** EGF stimulation and RAS co-IP analysis in HEK293 (wild-type *KRAS*) and **f** MIA PaCa-2 (*KRAS$^{G12C}$*) cells expressing FLAG-tagged *AGO2* (wild-type or Y393F). IP of endogenous AGO2 upon H$_2$O$_2$ treatment (4'), in H358 (**g**) and MIA PaCa-2 (**h**) cells harboring *KRAS* mutations, followed by immunoblot analysis of KRAS. **i** H$_2$O$_2$ treatment and KRAS-AGO2 co-IP analysis in MIA PaCa-2 (*KRAS$^{G12C}$*) cells expressing FLAG-tagged *AGO2* (wild-type or Y393F). Numbers on the left of the immunoblots in this panel indicate protein molecular weights in kDa. **j** Left panels, Representative images of single target (RAS or AGO2) and RAS-AGO2 interaction PLA in wild-type RAS expressing PC3 cells across the indicated cell culture conditions. Right panels, Representative images of PLA to detect RAS-AGO2 interaction in wild-type RAS expressing MCF-7 (panel I) and oncogenic KRAS expressing HCT116 (panel II) and H358 (panel III) cells grown in the indicated culture conditions. PLA signals appear as red dots around DAPI stained nuclei in blue. Scale bar, 50 μm.

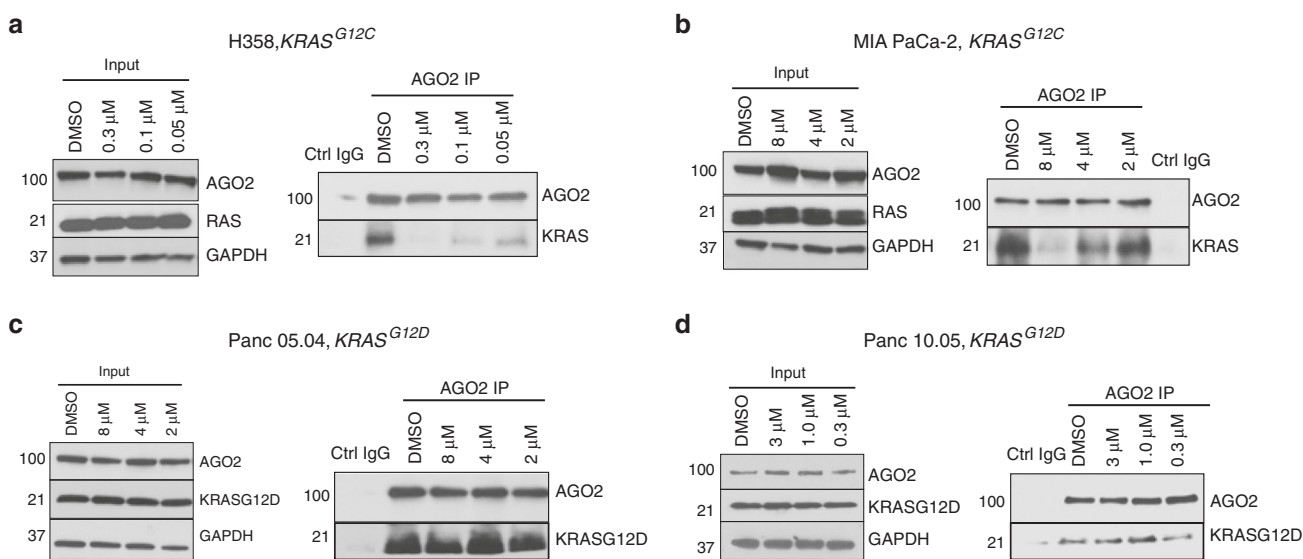

**Fig. 8 ARS-1620, a G12C-specific inhibitor, disrupts the KRAS$^{G12C}$-AGO2 interaction.** IP of endogenous AGO2 followed by immunoblot to detect KRAS in *KRAS$^{G12C}$* harboring (**a**) H358 and (**b**) MIA PaCa-2 cells treated with varying concentrations of ARS-1620 for three and nine hours, respectively. *KRAS$^{G12D}$* harboring (**c**) Panc 05.04 and (**d**) Panc 10.05 cells, respectively, treated with ARS-1620 for 24 h followed by AGO2 IP and immunoblot analysis of KRAS$^{G12D}$. For each cell line, input blots for AGO2 and RAS are shown.

different etiology for PDAC progression and limits our understanding of the requirement of AGO2 in advanced disease states.

This study also reveals how EGFR activation allows fine-tuning of RAS signaling by disrupting the membrane RAS-AGO2 association under conditions of stress (starvation or presence of oncogenic KRAS). Interestingly, phosphorylation of AGO2 by EGFR simultaneously inhibits the last step of microRNA biogenesis[12] and activates RAS at the plasma membrane[62]. We observe that EGF stimulation is sufficient to disrupt the wild-type KRAS-AGO2 interaction, but disruption of the oncogenic KRAS-AGO2 interaction requires activation of the growth receptor through inhibition of cytosolic tyrosine phosphatases. However, under both of these conditions, interaction with KRAS is dependent on the Y393 phosphorylation status of AGO2. In fact, inhibition of tyrosine phosphatase PTP1B through oxidative stress has been shown to lead to accumulation of AGO2$^{Y393}$ in oncogenic RAS-expressing cells, leading to senescence[63]. It is also intriguing that EGFR-mediated phosphorylation of AGO2$^{Y393}$ disrupts KRAS binding in a manner reminiscent of AGO2-Dicer binding and renders AGO2 incapable of RNAi activity[12,64]. While both KRAS and EGFR aberrations are mutually exclusive, both alter AGO2 RNAi function through distinct mechanisms[11].

Extending these observations to our mouse model, we predict that *AGO2* loss phenotypically mimics AGO2$^{Y393}$ phosphorylation. During EGFR-dependent PanIN development (Fig. 9), neither phosphorylated AGO2$^{Y393}$ nor the setting of *AGO2* loss permits KRAS association, and this stage is, therefore, independent of AGO2. On the other hand, during progression to PDAC that is less reliant on EGFR, accumulation of AGO2$^{Y393}$ in its non-phosphorylated form promotes KRAS binding, which is essential for PDAC progression. Mouse models expressing only the RAS binding-deficient form of AGO2 will validate the biological role of the KRAS-AGO2 interaction.

Finally, we find that the G12C covalent inhibitor disrupts the mutant KRAS$^{G12C}$-AGO2 interaction. The inhibitor is known to covalently attach to cysteine residues and make contacts at the Switch II pocket. This binding has been shown to interfere with nucleotide exchange because of reduced SOS1-KRAS$^{G12C}$ interaction[47]. Given that SOS and AGO2 compete for binding to the Switch II domain to regulate RAS-GTP levels, it is not surprising that AGO2 also fails to bind inhibitor-bound KRAS. Therefore, our data suggests that abrogation of the oncogenic KRAS-AGO2 association at the plasma membrane may represent a therapeutic opportunity for pancreatic cancer that warrants further investigation.

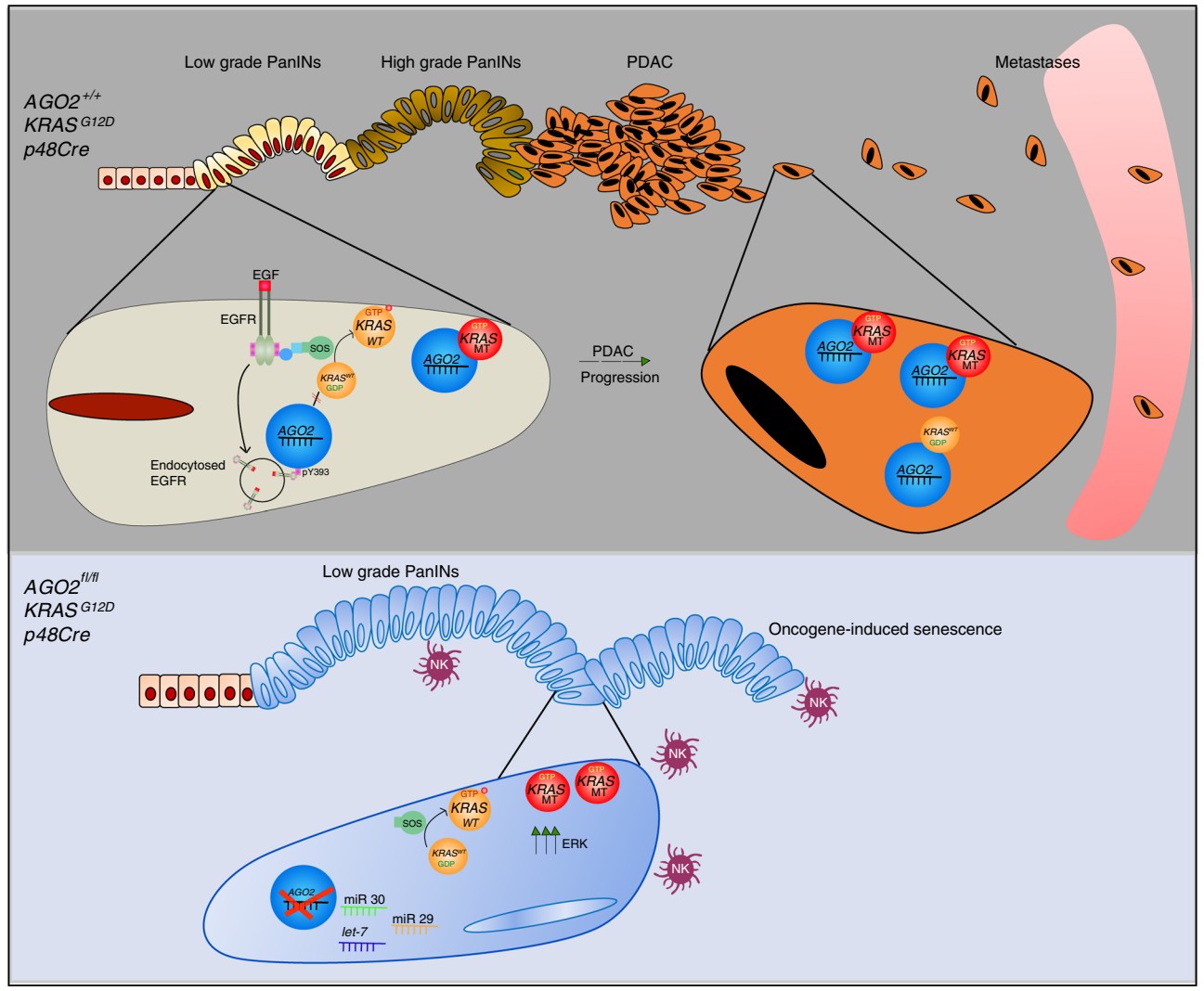

**Fig. 9 Schematic model showing the essential role of *AGO2* in PDAC progression.** Expression of *KRAS*$^{G12D}$ in normal pancreatic cells initiates low grade PanINs which progress to higher grade PanINs, PDAC, and metastases. PanIN formation requires EGFR that can phosphorylate AGO2 to disrupt the KRAS-AGO2 interaction and is, therefore, AGO2-independent. PDAC progression is associated with increased expression of KRAS and AGO2 at the membrane. *AGO2* ablation results in increased expression of microRNAs that regulate cell proliferation and senescence and also activates KRAS to promote oncogene-induced senescence. OIS due to *AGO2* loss prevents progression of low grade PanINs to PDAC and leads to infiltration by natural killer (NK) cells.

## Methods

**Mouse strains.** LSL-*KRAS*$^{G12D}$ (ref. [4]) (Kras $^{LSL-G12D}$) and *p48Cre* mice[65] were obtained from Marina Pasca di Magliano, University of Michigan. Conditionally floxed *AGO2* (ref. [13]) (*AGO2*$^{fl/fl}$) mice and *p53*$^{fl/fl}$ mice were purchased from Jackson labs (Bar Harbor, Maine). PCR genotyping for *KRAS*$^{G12D}$;*p48Cre, p53*$^{fl/+}$, and *AGO2* alleles, from DNA isolated from mouse tails, was performed using standard methodology. To generate experimental and control mice, *AGO2*$^{fl/fl}$ *p48Cre*, and *KRAS*$^{G12D}$ lines were intercrossed to generate *AGO2*$^{fl/+}$;*p48Cre* and *KRAS*$^{G12D}$;*p48Cre* mice. These two lines were then intercrossed to generate the *AGO2*$^{fl/fl}$;*KRAS*$^{G12D}$;*p48Cre* experimental mice. Given that mice were maintained on a mixed background, littermate controls were systematically used in all experiments (sex ratio per cohort was balanced). All animals were housed in a pathogen-free environment, and all procedures were performed in accordance with requirements of the University of Michigan IACUC. Cre activation in acinar cells of pancreata of mice with mutant *KRAS* alleles was validated by genotyping using the *KRAS*$^{G12D}$ conditional PCR detailed in Supplementary Table 3.

**Histology, immunohistochemistry, and IF.** Paraffin-embedded tissues from mice were processed using standard methodology. Details of the primary antibodies used for IHC are provided in Supplementary Table 1. Immunohistochemistry and IF staining were performed using standard techniques. For IF, slides were viewed using a Nikon 1A-B confocal microscope. To estimate co-localization of proteins, the Coloc2 program (ImageJ) was used to determine Pearson's coefficient. Cells within the PanIN/PDAC or metastatic regions (from mouse and human tissues), excluding the stromal compartment, were used to determine the extent of overlap.

In panels with normal tissue shown in Fig. 5, acinar cells were used for co-localization analyses. Average values over three different areas are shown.

**Proximity ligation assay (PLA).** Cell lines were cultured in 8-well chamber slides. After the indicated treatment/stimulation, cells were fixed with 4% paraformaldehyde and then permeabilized using 0.1% Tween. Subsequent PLA staining was performed as per the protocol provided by the manufacturer (DUOlink kit, Millipore/Sigma). Mouse RAS10 and rabbit AGO2 antibodies, validated in this study, were used at 1:250 dilution to detect signals either alone or in combination. Negative controls were performed using either single antibody (Supplementary Fig. 16a), Rasless MEFs (Supplementary Fig. 8d), or tissue lacking AGO2 (Fig. 5d). Images were obtained using the Nikon A1B inverted confocal microscope. For mouse tissue PLA, the paraffin-embedded sections were processed as for IF analysis. PLA was then performed using RAS10 or AGO2 antibodies, either alone or in combination, and imaged using the Nikon A1B confocal microscope.

**Human TMA analysis.** Pancreatic TMAs and frozen human tissue repositories were established by a pathologist (J.S.) and developed at the Tissue and Molecular Pathology Core in the Department of Pathology, University of Michigan, after IRB approval as described[66]. The Institutional Review Board at the University of Michigan approved the study (protocol number: HUM00098128). Patients with pancreas resections for pancreatitis, cystic neoplasms, or PDA from 2002 to 2015 at the University of Michigan Health System were included in the study. The electronic medical record was examined for clinical and demographic patient information. Date of surgery and date of last patient contact were recorded from the

electronic medical record. Deaths were confirmed from the Social Security Death Index. Clinical staging was analyzed using the American Joint Committee on Cancer 8th edition staging system. For patients who received neoadjuvant treatment, clinical stage was analyzed based on pre-treatment tumor size, while pathological parameters of tumor size, grade, lymph node status, and peripancreatic, duodenal, and common bile duct extension were analyzed based on the post-treatment surgical specimen. All hematoxylin and eosin (H&E) slides were reviewed and diagnoses confirmed by a gastrointestinal pathologist (J.S.), and corresponding areas were carefully selected and marked. Duplicated 1 mm diameter adjacent tissue cores from the same lesion in a total of 311 patient tissue samples were selectively punched/extracted and transferred to recipient tissue array blocks. Five TMAs were set up according to a standard protocol. H&E staining was performed on each TMA block using standard protocol, and unstained slides were prepared for immunohistochemical (IHC) staining and IHC scoring was performed by a pathologist (J.S.).

**RNA in situ hybridization (RNA-ISH)**. RNA-ISH was performed to detect *Kras* mRNA on formalin-fixed paraffin-embedded (FFPE) tissue sections using the RNAscope 2.5 HD Brown kit (Advanced Cell Diagnostics, Newark, CA) and target probes against mouse *Kras* (412491). *Mm-Ubc* (mouse ubiquitin C) and *DapB* (Bacillus bacterial dihydrodipicolinate reductase) were used as positive and negative controls, respectively. FFPE tissue sections were baked for 1 h at 60 °C, deparaffinized in xylene twice for 5 min each, and dehydrated in 100% ethanol twice for 1 min each, followed by air drying for 5 min. After hydrogen peroxide pre-treatment and target retrieval, tissue samples were permeabilized using Protease Plus and hybridized with the target probe in the HybEZ oven for 2 h at 40 °C. After two washes, the samples were processed for a series of signal amplification steps. Chromogenic detection was performed using DAB, counterstained with 50% Gill's Hematoxylin I (Fisher Scientific, Rochester, NY).

**Quantitative RT-PCR**. Pancreatic total RNA was isolated using the AllPrep DNA/RNA/miRNA Universal Kit (Qiagen). For quantitation of mRNA transcripts, RNA was extracted from the indicated samples, and cDNA was synthesized using the SuperScript III System according to the manufacturer's instructions (Invitrogen). Quantitative RT-PCR was conducted using primers detailed in Supplementary Table 3 with SYBR Green Master Mix (Applied Biosystems) on the StepOne Real-Time PCR System (Applied Biosystems). Relative mRNA levels of the transcripts were normalized to the expression of the housekeeping gene *GAPDH*.

**MiRNA expression profiles using qPCR of mouse miRnome panels**. Pancreatic total RNA was isolated using AllPrep DNA/RNA/miRNA Universal Kit (Qiagen). Five nanograms of total RNA from each sample was converted into cDNA using miRCURY™ LNA™ Universal RT microRNA PCR Universal cDNA Synthesis Kit II. Quantitative micro RT-PCR was performed using exiLENT SYBR Green master mix with microRNA ready to use PCR mix, Mouse&Rat panel I, V4.M (Exiqon, Cat # 203713) on ABI 7900HT Fast Real-time PCR system (Applied Biosystems). Data were analyzed using GenEX ver 6 software.

**Transcriptome analysis**. mRNA was quantitated on the Illumina platform using the Riboerase library preparation protocol. Transcriptome data processing and quality control were performed using RSeQC package. Sequencing alignment was performed using splice aware aligner STAR with two pass alignment option using mm10 reference build. featureCounts from Rsubread package was used to get the count matrix for expression quantification. R-package edgeR DGEList object was used to import, organize, filter, and normalize the data by the method of trimmed mean of M-values (TMM)[67] using the calcNormFactors. This was followed by limma[68] and voom[69] analyses where default settings for the "voom","lmFit", "eBayes", and "topTable" functions were used to assess differential gene expression. Finally, the fgsea package was used to perform gene set enrichment analysis. Software version details are: edgeR_3.28.0, limma_3.42.0, Rsubread_2.0.0 (featureCounts), RSeQC-2.6.4, STAR-2.7.3a.

**Pancreatic tissue lysates and immunoblot analysis**. Pancreata obtained from mice were homogenized in $Mg^{2+}$-containing lysis buffer. Clear lysates were separated using SDS-PAGE and processed for immunoblot analysis using standard methods. Primary antibodies used in the study are indicated in Supplementary Table 1. Particularly, Ras antibodies validated in a recent study[29] are also indicated. IMAGEJ (ImageJ-win64) was downloaded from https://imagej.net/Fiji/Downloads.

**Isolation of pancreatic ductal organoids**. Pancreatic ducts were isolated from the pancreas of 12-week-old $KRAS^{G12D}$;*p48Cre* and $AGO2^{fl/fl}$;$KRAS^{G12D}$;*p48Cre* mice[39] by enzymatic digestion with 0.012% (w/v) collagenase XI (Sigma) and 0.012% (w/v) dispase (GIBCO) in DMEM media containing 1% FBS (GIBCO). Organoids were seeded in growth factor-reduced (GFR) Matrigel (BD). Organoid culture medium consisted of Advanced DMEM/F12 (Invitrogen), B27 (Invitrogen), 1.25 mM N-Acetylcysteine (Sigma), 10 nM gastrin (Sigma) and the following growth factors: 50 ng/ml EGF (Peprotech), 10% RSPO1-conditioned media (prepared in-house), 10% Noggin-conditioned media (prepared in-house[70]), 100 ng/ml FGF10

(Peprotech), and 10 mM nicotinamide (Sigma). For experiments, organoids were released from the Matrigel, mechanically disrupted into small fragments, and plated in fresh Matrigel. To enrich pancreatic ducts from tissues expressing oncogenic KRAS, no EGF was added to the base medium. Organoids were passaged at a 1:4–1:6 split ratio once per week for at least 9 weeks to enrich $KRAS^{G12D}$ expressing organoids.

To carry out erlotinib treatment, organoid cultures were dissociated and split into equal parts and cultured in Matrigel for the indicated times. To collect untreated and treated samples, organoids in Matrigel were enzymatically dissociated, collected, and washed prior to protein isolation.

**RAS-GTP analysis**. 300–500 micrograms of indicated protein lysates were prepared from pancreatic ductal organoids or cell lines using $Mg^{2+}$-containing lysis buffer. RAF1-RBD agarose beads (Millipore) were used to pull down activated RAS-GTP[11]. The beads were washed and separated using SDS-PAGE and immunoblotted for the indicated proteins.

**Plasmids**. Full-length FH-*AGO2* constructs were obtained from Addgene (pIR-ESneo-FLAG/HA-AGO2 10822, PI:Thomas Tuschl). $AGO2^{Y393F}$ mutant construct was generated using the QuikChange II XL Site-Directed Mutagenesis Kit (Agilent) from the FH-*AGO2* plasmid described above using the primers detailed in Supplementary Table 3. DNA sequences were confirmed using Sanger sequencing at the University of Michigan Sequencing Core.

**In vitro assay to measure KRAS-GTP levels**. Purified catalytic domains of SOS1 (Cytoskeleton) and NF1 (Creative Biomart) were used to carry out nucleotide exchange and GTPase activity. Full-length KRAS and $KRAS^{G12V}$ were purified using bacterial expression at the University of Michigan Proteomic Core. AGO2 was purchased from Sino Biologics. Halo tag protein was obtained from Promega. Purified components were added as indicated, and KRAS-GTP levels were estimated using the GTPase-GLO assay from Promega, following the manufacturer's instructions.

**Cell culture, transfection, and EGF stimulation**. All cell lines (detailed in Supplementary Table 4) were obtained from the American Type Culture Collection (ATCC) or as indicated. Cells were cultured following ATCC culture methods in media supplemented with the corresponding serum and antibiotics. Additionally, cells were routinely genotyped and tested bi-weekly for mycoplasma contamination. Only cells with the correct genotype and that were mycoplasma free were used for the experiments. For EGF stimulation, cells were grown to approximately 80% confluence and washed with PBS three times. Cells were incubated overnight (16 h) in serum free media. EGF stimulation was performed for 5 min with 100 ng/mL of EGF (Gibco) at 37 °C. After stimulation, cells were washed and protein lysates were prepared in K Buffer lysis buffer. For tyrosine kinase inhibition, cells were pretreated with 15 μM of erlotinib for 1 h prior to EGF stimulation, as described above.

HEK293 or MIA PaCa-2 cells were transfected with different *AGO2* constructs using Fugene HD (Promega) or Lipofectamine 3000 (Invitrogen) according to the manufacturer's protocols. For EGFR stimulation with transient *AGO2* construct overexpression, cells were transfected ~16 h prior to overnight serum starvation and EGF stimulation.

RASless MEFs were a kind gift from the RAS Initiative. Details of how these cells were developed and their growth characteristics can be found at https://www.cancer.gov/research/key-initiatives/ras/ras-central/blog/2017/rasless-mefs-drug-screens.

**Immunoprecipitation (IP) analysis**. For immunoprecipitation analysis, protein lysates were prepared in K Buffer (20 mM Tris pH 7.0, 5 mM EDTA, 150 mM NaCl, 1% Triton X100, 1 mM DTT, phosphatase inhibitors, and protease inhibitors). Typically,150–200 μg of protein lysates (RAS10 IP: 150 μg; AGO2 IP: 200 μg; KRAS IP: 150 μg) were pre-cleared with 10 μl of Protein A/G agarose beads (Santa Cruz) for 1 h. Pre-cleared lysates were incubated with 5–10 μg of the indicated primary antibodies targeting the protein of interest or with corresponding isotype controls overnight at 4 °C. Thirty microliters of Protein A/G beads were then added to immune complexes and incubated for 1–3 h at 4 °C, spun, and washed in 150–300 mM NaCl containing K-buffer prior to separation of immunoprecipitates by SDS-PAGE. To determine the varying levels of KRAS expressed in different cells lines (with or without EGF stimulation), shown in Fig. 7, pan RAS10 antibody was used for immunoprecipitation followed by immunoblot analysis using KRAS-specific SC-30 antibody.

**β-galactosidase assay**. β-galactosidase staining was performed using the Senescence β-Galactosidase Staining Kit #9860 (Cell Signaling) on 10 μM-thick frozen sections of mouse pancreas, as per the manufacturer's protocol.

**Statistics and reproducibility**. Reproducibility of results were ensured by (1) involving multiple members of the team to collect data, (2) analyzing pathologies with two independent pathologists, and (3) repeating cell line-based experiments at least twice. Many experiments were repeated by multiple members of the group.

The data are representative of multiple biological replicates, and the number of times each of the individual data was repeated with similar results is indicated below.

Main Figures

Two: 5c, 5f, 6d, 6e, 6f, 7g, 7h, 7j, 8c, 8d

Three: 1b, 2c, 2d, 5a, 5g, 6g, 6h, 7a, 7b, 7d, 7i, 8a, 8b

Four: 2f, 7c, 7e, 7f

Six: 6i, 6j, 6k

Eight: 3c

Ten: 4a, 1c, 5b, 4d, 5e

Also note that for Fig. 2a, the image is representative of data collected in Fig. 2b, and representative images for Fig. 4b were obtained by analysis of six PanINs, six PDAC, and two metastatic tissues.

Supplementary Figures

Two: 1a, 2, 7b, 7c, 7d, 8b, 8c, 8d,13c, 14c, 15a, 15b, 15c, 16a, 16b

Three: 7e, 9a, 14a, 14b

Four: 6, 9c, 13a,

Six: 3a, 4a, 5, 8a

Micrographs shown in 4b represent each of the abnormal pathologies observed.

**Reporting summary**. Further information on research design is available in the Nature Research Reporting Summary linked to this article.

## Data availability

All data generated or analyzed during this study are included in this article. This includes raw data for the immunoblot (Main Figs. 2c; 6c, e, f, g, h; 7a, b, c, d, e, f, g, h, i; 8a, b, c, d; and Supplementary Figs. 2; 7b, c; 12a; 15a, b, c) and microRNA analyses (Fig. 6a, Supplementary Fig. 10a, b). A reporting summary for this article is available as a Supplementary Information file. The RNA sequencing data (for Fig. 6b and Supplementary Fig. 11) have been deposited in GEO under the accession number GSE147781.

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

## Acknowledgements

We thank Kristin M. Juckette, Fengyun Su, Grace Tsaloff, Lisha Wang, and Javed Siddiqui for their assistance with experimental work. We thank Mandy Davis and Marta Hernadi-Muller for their help with processing paraffin-embedded slides. We also thank Saravana Dhanasekaran, Miriam Gandham, and Markus Eberl for technical assistance. We thank members of the RAS Initiative, Dhirendra Simanshu (for discussions) and Jim Hartley (for providing RASless MEFs). We are also thankful for RAS/AGO2-related discussions with Phillip Sharp, Eric Fearon, and David Bartel. We acknowledge Sisi Gao for help in manuscript preparation. S.S. is a Genentech Fellow. J.S. is supported by NCI K08CA234222. H.C.C. is supported by NIH Grants U01 CA224145 and DOD CA170568. A.M.C. is a National Cancer Institute Outstanding Investigator (R35CA231996), Howard Hughes Medical Institute Investigator, A. Alfred Taubman Scholar, and American Cancer Society Professor.

## Author contributions

Mouse experimental data were generated by J.C.T., S.C., A.G., A.X., V.L.D., and S.S. Contributions to other experimental data were made by S.S., R.F.S, V.L.D., S.Z.-W., S.E., X.C., M.M., S.P., I.J.A., and C.K.-S. R.M. and J. S. coordinated the pathology assessment. J.S. provided the human TMA and performed IHC scoring. X.W. performed RNA ISH. J.W. and J.T. supported work on AGO2 phosphorylation. P.V. and Y.Z. provided bioinformatics support. S.J.E. and X.C. helped with project management. H.C.C. supervised the pancreatitis experiments. S.S. and A.M.C. jointly conceived the study. S.S., C.K.-S., and A.M.C. wrote the manuscript. Funding and overall supervision of the study was provided by A.M.C.

## Competing interests

The authors declare no competing interests.
