## [Peer Review File · Nature Communications]

Reviewers' comments:

Reviewer #1 (Remarks to the Author); expert in PDAC, EGFR, KRAS:

The authors previously identified AGO2 as a KRAS interacting protein and important for the growth of KRAS mutant cancer cells. In the present study, they utilized mouse models to address the role of AGO2 in KRAS driven pancreatic cancer development. An AGO2 deficiency in the pancreas did not impair normal pancreas development or KRAS driven formation of early stage panIN lesions. However, AGO2 deficiency prevented progression to metastatic pancreatic cancer and prolonged survival. The basis for AGO2 deficiency impaired progression induction of senescence associated with elevated ERK activation. Concurrent TP53 loss overcame the AGO2 deficiency and resulted in pancreatic cancer development. Next, a series of studies evaluated AGO2 protein expression and interaction with KRAS. Elevated AGO2 protein levels were seen in both mouse and human pancreatic tumors. Furthermore, an increased membrane colocalization of RAS and AGO2 was found with pancreatic cancer progression. AGO2 deficiency was also associated with increased phosphorylation/activation of the EGFR and increased WT RAS-GTP formation. Further analyses of AGO2-KRAS interaction was done in either WT or mutant KRAS cell lines. Upon EGF stimulation and EGFR activation, AGO2-RAS interaction was reduced with WT but not mutant KRAS, and disruption of interaction was dependent on AGO2 phosphorylation.

This study is an interesting extension of the authors' recent characterization of AGO2 interaction with RAS. The conclusions are generally supported by the data shown. The study characterizes additional roles by which AGO2 may impact RAS function, signaling and supporting cancer growth. However, two limitations with the study dampen enthusiasm for recommending publication. First, some of the findings (as cited below) in this current study are seemingly contradictory to findings made previously by the author regarding AGO2 interaction with KRAS. Second, several interesting observations are made, but not mechanistically explained, to account for findings made. These observations are mentioned below. A revised study with additional experimental analyses that address these two broad issues can be reconsidered for publication,

Specific concerns:

(Fig. 2) – The changes in KRAS effector signaling seen with AGO2 deficiency seem at odds to the author's previous findings that AGO2 binds to the switch 2 domain of KRAS, and hence, competes with PI3K but not RAF effector interaction. Why would ERK signaling but not PI3K signaling increase if KRAS is no longer occupied with AGO2 association? Is this increased ERK activation not due to mutant KRAS and instead to EGFR activation of WT RAS as their later data suggests? If yes, this would also be inconsistent with the author's previous observation that mutant KRAS levels are decreased by an AGO2 deficiency.

(Fig. 3) – If loss of TP53 negates a requirement for AGO2 for tumor progression, does AGO2 suppression in malignant pancreatic cancer lines impair growth only in TP53 WT but not mutant cell lines? Does loss of TP53 not alter any of the AGO2 changes seen, such as AGO2 increased expression and colocalization with RAS with progression? While this is a very interesting finding, why a TP53 deficiency overcomes the loss of AGO2 is not well-evaluated mechanistically regarding AGO2 function.

(Fig. 5) – the detection of "membranous RAS expression" was stated to be "indicative of activated RAS" (Fig. 5b); this is not accurate, since activated RAS is defined as the GTP-bound protein. The increased RAS membrane staining in Fig. 5e was also described as "an increased level of activated RAS". Further, since the antibody used is a pan-RAS antibody, it is detecting both mutant and wild type RAS proteins. Since both RAS and AGO2 staining was plasma membrane associated, the authors conclude that RAS and AGO2 "colocalize". This conclusion is overstated, since co-staining patterns alone is not sufficient evidence to conclude co-localization.

(Fig. 6a,c) – In their previous studies with cancer cell lines, they observed significant reduction in mutant KRAS protein levels with AGO2 KD. However, in PanIN lesions, they found no change in mutant KRAS levels, and furthermore, increased total RAS levels. What is the basis for these different observations?

(Fig. 6c) – Data in this figure show that there is increased total RAS-GTP, and it is assumed that this is due to increased WT and not mutant RAS. Use of RAS isoform specific antibodies should be done to determine what RAS isoform, WT or mutant, is activated by loss of AGO2. Also, the assumption is that this increase is due to increased EGFR signaling – treatment of organoids with an EGFR inhibitor should be done to verify the role of EGFR in this activation.

(Fig. 6d) – In MEFs they show that AGO2 loss is associated with increased EGFR activity and RAS-GTP formation – is pERK but not pAKT increased as seen in mice with an AGO2 deficiency? Ectopic reexpression of AGO2 reversed these activities. The authors previously described an AGO2 mutant that cannot bind RAS, AGO2 K112A. Showing this mutant cannot reverse these activities would support the importance of AGO2 interaction with WT RAS in causing these signaling changes.

(Fig. 7a) – While the data do support the author's conclusion that EGF stimulation impairs AGO2 association with WT but not mutant KRAS, no data are presented to provide an explanation for this difference. Since the authors previously found that the AGO2-RAS interaction is independent of the nucleotide bound, the relative GTP-bound state of mutant versus WT KRAS should not be a factor.

(Fig. 7b) – The requirement for EGF stimulation to cause increased pERK levels in the KRAS mutant cancer lines is puzzling, since mutant KRAS alone, independent of upstream RTK signaling, should be capable of binding and activating RAF.

(Fig. 8) – The ability of a G12C selective inhibitor to disrupt mutant KRAS-AGO1 interaction is confusing regarding what is known about AGO2-RAS interaction. The G12C inhibitor covalently modifies GDP-bound KRAS G12C, preventing formation of the GTP-bound state, leading to loss of effector binding. Since AGO2-RAS interaction is not nucleotide dependent, is this disruption due to loss of PI3K binding, or because the inhibitor-modified KRAS protein can no longer bind AGO2?

(Fig. 9) – This model, while useful, is not necessary, since the concept is not complex and easily understandable by words in the text. However, some of the mechanistic issues that regulate AGO2-KRAS interaction, such as the role of EGFR signaling, would be better explained by a model figure. A model showing how EGFR activation can disrupt WT but not mutant KRAS association, and how an AGO2-RAS association alters RAS function, would be more informative.

Reviewer #2 (Remarks to the Author); expert in AGO2 and RNAi machinery:

In this manuscript, Shankar et al. describe an unexpected role for AGO2 in KRASG12D-driven pancreatic cancer mouse model. They find that AGO2 is dispensable for the early stage of cancer development but is required for the development of PDAC and metastasis. They demonstrate that AGO2 expression is required to block oncogene-induced senescence (OIS), which can be bypassed by abolishing p53 regulation. The authors propose that KRAS-AGO2 interaction is critical for this unconventional role of AGO2 in repressing OIS.

Overall, this study is solid and interesting. By mouse model studies, authors provide compelling evidence that AGO2 is required for PanINs to overcome OIS and progress to PDAC. The fact that AGO2 expression is elevated in human PDAC samples further supports the role of AGO2 plays in

pancreatic cancer development.

However, the mechanistic part of this study is relatively weak. The same group has reported previously that KRAS-AGO2 interaction enhances cellular transformation based on cell line studies (Shankar et al, Cell Rep, 2016). Despite a fair amount of work, the current study does not bring much new to the underlying mechanism. The observation that phosphorylation of AGO2 by EGFR disrupts the interaction of WT KRAS, but not mutant KRAS, with AGO2, although by itself intriguing, does not contribute much towards the understanding of the most critical question: How does AGO2 overcome the OIS block? I would encourage the authors to address or at least discuss the following points/questions:

1. RAS-AGO2 interaction correlates with PDAC. It would greatly strengthen their conclusions if the authors can establish the causality in their mouse models.
2. The inhibition of OIS is a result of modulating the function of AGO2, KRAS/KRAS-mut, or both? Specifically, the role of AGO2 in this process is microRNA dependent or independent? How is the function of KRAS/KRAS-mut impacted by the AGO2 association? Transcriptome analyses by RNA-Seq as well as the profiling of miRNAs by miRNA-Seq should provide insights.

Reviewer #3 (Remarks to the Author); expert in oncogene-induced senescence:

Manuscript: #NCOMMS-19-25257-T

Title: An essential role for argonaute 2 in EGFR-KRAS signaling in pancreatic cancer development

Authors: Shankar et al.

In the manuscript " An essential role for argonaute 2 in EGFR-KRAS signaling in pancreatic cancer development" (#NCOMMS-19-25257-T), Shankar and colleagues delineate a complex signal transduction pathway whereby epidermal growth factor receptor (EGFR) signaling provokes the dissociation of a supramolecular complex between wild-type (but not mutant) KRAS and argonaute 2 (AGO2), which is relevant for pancreatic oncogenesis driven by KRASG12D, as the latter occurs in an initial EGFR-dependent phase, and a subsequent EGFR-independent phase.

The findings by Shankar et al. are novel and the study is globally well performed and in line with the scope of Nature Communications. The manuscript is well written and figures are clear, although not particularly well organized (and in some instances contains excessive/inaccurate statements). The technological portfolio employed by the authors to demonstrate their hypotheses is not particularly variegated (largely, imaging often not combined with quantitative analysis, immunoblot and co-immunoprecipitation), but globally serves the purpose. In summary, I believe this work is a good candidate for publication in Nature Communications, once the following points have been properly addressed.

Major issues:

1. Little has been done about the cellular mechanisms whereby the AGO2-KRAS complex impact on oncogenesis and tumor progression. The data are strikingly similar to work from Eileen White demonstrating that defects in Atg7 favor early oncogenesis in the lung and yet prevent progression to advanced disease stage. Given the results with serum starvation and the well described role EGFR and KRAS in autophagy inhibition, investigating the impact of autophagy would be a great addition here.
2. Along similar lines, it is surprising that the authors did not investigate whether the control of lesions undergoing senescence in the absence of AGO2 is mediated by the immune system. Senescence surveillance by NK cells is well established and could play a major role in this model.
3. Do similar observations apply to other types of cancer driven by KRAS mutations, like lung carcinoma? Understanding the extension of the phenomenon would be important

4. A transcriptional comparison of AGO2 competent and incompetent lesions is needed to obtain additional insights into the underlying mechanisms
5. Fig. 6A,B. Total EGFR levels change considerably in immunoblot, not in IHC. How do the authors explain this? In general, many immunoblots require densitometry and quantitation.
6. Fig 7G is missing, which prevents evaluation of the corresponding text.

Minor issues:

1. Some statements and data interpretation are inaccurate or excessive and need to be corrected or toned down. As a few examples: line 83 "closely mimics" is excessive, given the exclusivity of KRAS and EGFR mutations in human but the need for EGFR signaling in the model; line 112 "faithfully mimicking", same problem; line 386, this is not true, it can reflect a distant binding and conformational change.
 2. Fig. 5E, the authors points to increase levels of activated KRAS between PnaINs and PDACs, which is not completely supported by non-quantitative imaging
 3. Figure 3A. The authors conclude that p53 knockout can abolish the survival advantage obtained by AGO2 knockout (Fig. 1F), but the two scenarios are difficult to compare as median survival is completely different. This should be commented upon: the disease in the absence of p53 is a completely different one
 4. Figure 3C. AGO2 positive islands are no longer visible in the Ago2 KO setting, at odds with Fig 1B.
 5. Figure 4B, the authors may want to explore TCGA data in support of their hypothesis.
 6. It would have been preferable to have composed figures rather than sparse panels.
- END

Reviewers' comments:

Reviewer #1 (Remarks to the Author); expert in PDAC, EGFR, KRAS:

The authors previously identified AGO2 as a KRAS interacting protein and important for the growth of KRAS mutant cancer cells. In the present study, they utilized mouse models to address the role of AGO2 in KRAS driven pancreatic cancer development. An AGO2 deficiency in the pancreas did not impair normal pancreas development or KRAS driven formation of early stage panIN lesions. However, AGO2 deficiency prevented progression to metastatic pancreatic cancer and prolonged survival. The basis for AGO2 deficiency impaired progression induction of senescence associated with elevated ERK activation. Concurrent TP53 loss overcame the AGO2 deficiency and resulted in pancreatic cancer development. Next, a series of studies evaluated AGO2 protein expression and interaction with KRAS. Elevated AGO2 protein levels were seen in both mouse and human pancreatic tumors. Furthermore, an increased membrane colocalization of RAS and AGO2 was found with pancreatic cancer progression. AGO2 deficiency was also associated with increased phosphorylation/activation of the EGFR and increased WT RAS-GTP formation. Further analyses of AGO2-KRAS interaction was done in either WT or mutant KRAS cell lines. Upon EGF stimulation and EGFR activation, AGO2-RAS interaction was reduced with WT but not mutant KRAS, and disruption of interaction was dependent on AGO2 phosphorylation.

This study is an interesting extension of the authors' recent characterization of AGO2 interaction with RAS. The conclusions are generally supported by the data shown. The study characterizes additional roles by which AGO2 may impact RAS function, signaling and supporting cancer growth. However, two limitations with the study dampen enthusiasm for recommending publication. First, some of the findings (as cited below) in this current study are seemingly contradictory to findings made previously by the author regarding AGO2 interaction with KRAS. Second, several interesting observations are made, but not mechanistically explained, to account for findings made. These observations are mentioned below. A revised study with additional experimental analyses that address these two broad issues can be reconsidered for publication,

RESPONSE: We thank the reviewer for their comments and in-depth review. In the revised manuscript, we present additional data to address how AGO2 regulates RAS function and provide mechanistic details that explain the OIS due to AGO2 loss. We believe that the new data strengthen our previous observations and also explain the striking phenotype we observe in the mouse model with AGO2 loss.

Specific concerns:

(Fig. 2) – The changes in KRAS effector signaling seen with AGO2 deficiency seem at odds to the author's previous findings that AGO2 binds to the switch 2 domain of KRAS, and hence, competes with PI3K but not RAF effector interaction.

RESPONSE: In our previous study, we focused on PI3K signaling since Akt activation was significantly regulated in both AGO2 knockdown (H358 cells) and AGO2 overexpression (NIH3T3 cells) models.

However, in the same report, we did observe slight elevation of pERK signal in NIH3T3 AGO2^{-/-} cells expressing oncogenic KRAS (Cell Reports, 2016, 14, 1448–1461: Fig. S7D)¹.

We also reported that AGO2 knockdown in H358 cells leads to a general loss of multiple signaling pathways, including pERK (Cell Reports, 2016, 14, 1448–1461: Fig. S5 C, D and E)¹. This suggests that AGO2 ablation can alter multiple RAS effector pathways in a cell line-dependent manner.

It has been demonstrated that cell line models exhibit distinct KRAS dependencies on various effector pathways for growth, proliferation, and cellular transformation². Specifically, these could be attributed to differences in growth factor signaling through the wild-type RAS isoforms present in oncogenic KRAS-expressing cells³. What remains consistent across various models is the requirement for AGO2 for oncogenic KRAS-driven cellular transformation.

Why would ERK signaling but not PI3K signaling increase if KRAS is no longer occupied with AGO2 association?

RESPONSE: We acknowledge this needs further clarification, and to address this, we have added new experiments in the revised manuscript. When KRAS is no longer bound to AGO2, it results in increased KRAS-GTP levels (Fig. 6E-H in revision). As noted in the preceding response, increased KRAS activation can stimulate multiple pathways, yet in the models of AGO2 loss described in this study (pancreatic tissue, organoids, and MEFs), there was a preferential increase in ERK activation (Fig. 2C, 6E, and 6G).

To experimentally address how KRAS-GTP levels could increase in the absence of AGO2, we performed biochemical assays that monitor the levels of KRAS-GTP in the presence of various RAS regulators. Specifically, using purified catalytic domains of Neurofibromin 1-GAP (GTPase activating protein) and SOS1-GEF (guanine nucleotide exchange factor), we measured KRAS-GTP levels in the presence or absence of AGO2. As shown in Fig. 6I, AGO2 had no effect on the intrinsic GTPase activity of KRAS and did not alter NF1-GAP activity on KRAS. However, SOS1-mediated nucleotide exchange on wild-type RAS was significantly reduced in the presence of AGO2 (Fig. 6J) in a dose-dependent manner (Fig. 6K). In similar assays, oncogenic forms of KRAS were resistant to both GAP and GEF activity.

Given that both SOS and AGO2 compete for binding to the KRAS Switch II domain through Y64, AGO2 interaction limits wild-type RAS activation by preventing SOS binding. Indeed, RAS activation observed in AGO2^{-/-} MEFs could be diminished by reconstitution of wild-type AGO2 but not with RAS binding-deficient AGO2 (Fig. 6H). Downstream RAS signaling through ERK activation tracked with RAS-GTP levels. Together, these data suggest that the KRAS-AGO2 interaction competes with GEFs to limit wild-type RAS activation and thereby reduce RAS effector function.

Is this increased ERK activation not due to mutant KRAS and instead to EGFR activation of WT RAS as their later data suggests?

RESPONSE: At the 400-day time point, when the ERK activation is observed, EGFR levels were significantly reduced (Supplementary Fig. 13A). Therefore, ERK activation at this stage could be attributed to:

- a) hyperactive mutant RAS at the membrane and/or
- b) wild-type RAS activation due to AGO2 loss, through increased SOS-GEF nucleotide exchange (Fig. 6E, F, J, and K).

Given the lack of antibodies that can distinguish between different forms of RAS by IHC/IF, we are unable to discern the specific contributions of wild-type and mutant RAS present at the membrane in the PanIN lesions of mice lacking AGO2 (Supplementary Fig. 8A).

If yes, this would also be inconsistent with the author's previous observation that mutant KRAS levels are decreased by an AGO2 deficiency.

RESPONSE: In the mouse model, AGO2 loss did not result in decreased mutant KRAS expression (Fig. 6C, E and Supplementary Fig. 12). We agree that this is in contrast to our earlier experiments using cancer cell lines. This discrepancy could be attributed to the interplay of other accumulated mutations (besides KRAS) that are abundant in these cell lines and context dependencies due to cell culture conditions. *Indeed, cell line-based limitations were what prompted us to study the effects of AGO2 loss in the pancreatic cancer GEMM.*

(Fig. 3) – If loss of TP53 negates a requirement for AGO2 for tumor progression, does AGO2 suppression in malignant pancreatic cancer lines impair growth only in TP53 WT but not mutant cell lines?

RESPONSE: To address if AGO2 loss in p53-sufficient cells impairs cell proliferation, we performed CRISPR-mediated knockout of AGO2 in pancreatic cell lines harboring KRAS^{G12D} with and without p53 mutation⁴. As shown below in the Rebuttal Fig. 1A and B, AGO2 knockout in both p53 mutant (Panc 10.05) and p53 wild-type (SW1990) pancreatic cells leads to only a modest reduction in cell proliferation rates. The lack of AGO2 dependency in p53-sufficient SW1990 cells could be due to loss of other important tumor suppressors like CDKN2A that control senescence. As described earlier¹, not all KRAS mutated cells are dependent on AGO2 for cell proliferation. Our study using the GEMM has therefore proved to be critical for our understanding of the role of AGO2 to establish

Rebuttal Figure 1: AGO2 knockout alone has minimal effect on growth rates in select pancreatic cancer cell lines. Proliferation rates in Panc10.05 (A) and SW1990 (B) cells with CRISPR/Cas9-mediated AGO2 knockout.

PDAC.

Does loss of TP53 not alter any of the AGO2 changes seen, such as AGO2 increased expression and colocalization with RAS with progression?

RESPONSE: In the revised manuscript, we have provided data to address this query. As shown in Supplementary Fig. 9B, AGO2 expression increases during PDAC progression, even in the KPC model. This was also associated with increased membrane colocalization with RAS (Supplementary Fig. 9C). This pattern of staining is therefore consistent across both the mouse models expressing oncogenic KRAS and, more importantly, similar to the staining pattern within human pancreatic tissues with PDAC.

While this is a very interesting finding, why a TP53 deficiency overcomes the loss of AGO2 is not well-evaluated mechanistically regarding AGO2 function.

RESPONSE: The revised manuscript provides new mechanistic insights related to the microRNA functions of AGO2 that play a role in the OIS. We performed microRNA and transcriptomic analysis of pancreatic tissues obtained from *AGO2^{+/-};KRAS^{G12D};p48Cre* and *AGO2^{fl/fl};KRAS^{G12D};p48Cre* mice. As shown in Fig. 6A and Supplementary Fig. 10A, a set of microRNAs were differentially regulated between the two groups. Strikingly, the let-7, miR-29, and miR-30 families were significantly downregulated in the presence of AGO2 compared to pancreata lacking AGO2. Interestingly, expression of Rb-regulated miR-29 and miR-30 family members has been shown to induce senescence^{5,6}. Therefore, reduced expression of these microRNAs in the pancreatic tissue expressing AGO2 could prevent senescence and allow progression into PDAC. This was evident in the transcriptomic profiles of AGO2-null pancreatic tissues which showed reduced expression of E2F target genes, G2/M checkpoint, and oncogenic RAS signaling (Fig. 6B and Supplementary Fig. 11).

To address how p53 loss overcomes this phenotype, we quantitated the expression of these OIS-associated microRNAs in the KPC model. Downregulation of this family of microRNAs was apparent in both genotypes independent of AGO2 (Supplementary Fig. 10B), thus offering an explanation for how AGO2 loss in the setting of TP53 deficiency overcomes the OIS phenotype.

(Fig. 5) – the detection of “membranous RAS expression” was stated to be “indicative of activated RAS” (Fig. 5b); this is not accurate, since activated RAS is defined as the GTP-bound protein. The increased RAS membrane staining in Fig. 5e was also described as “an increased level of activated RAS”. Further, since the antibody used is a pan-RAS antibody, it is detecting both mutant and wild type RAS proteins. Since both RAS and AGO2 staining was plasma membrane associated, the authors conclude that RAS and AGO2 “colocalize”. This conclusion is overstated, since co-staining patterns alone is not sufficient evidence to conclude colocalization.

RESPONSE: We agree with the reviewer that it is inaccurate to indicate membranous RAS as the activated form of RAS. In the revised manuscript, we have made the necessary changes to reflect that. We have also limited ‘RAS/AGO2 co-localization’ to the proximity ligation assay (PLA) since the PLA assay is designed to detect protein-protein interactions.

(Fig. 6a,c) – In their previous studies with cancer cell lines, they observed significant reduction in mutant KRAS protein levels with AGO2 KD. However, in PanIN lesions, they found no change in mutant KRAS levels, and furthermore, increased total RAS levels. What is the basis for these different observations?

RESPONSE: The reviewer raises an important question. We would like to reiterate that differences in the model systems contribute to the apparent discrepancy. The mouse model allows us to evaluate the mechanisms of progression from benign PanINs (dependent on wild-type RAS/EGFR) to PDAC, but the cancer cell lines reflect isolated cancer cells grown in culture and represent the terminal stages of KRAS-driven disease (with a number of accumulated mutations) and different requirements for wild-type RAS/EGFR.

Regarding the increased expression of wild-type RAS seen in Fig. 6C, we present data that suggests that it is in its GTP-bound activated form (Fig. 6E-F). It is possible that effector proteins that bind the activated wild-type RAS at the membrane protect from the cytosolic degradation machinery.

(Fig. 6c) – Data in this figure show that there is increased total RAS-GTP, and it is assumed that this is due to increased WT and not mutant RAS. Use of RAS isoform specific antibodies should be done to determine what RAS isoform, WT or mutant, is activated by loss of AGO2. Also, the assumption is that this increase is due to increased EGFR signaling – treatment of organoids with an EGFR inhibitor should be done to verify the role of EGFR in this activation.

RESPONSE: We thank the reviewer for this comment and for the suggestion to explore the organoid models further. We believe that the new data provides strong experimental evidence in support of our hypothesis.

To determine the specific RAS isoform activated in AGO2-null organoids, we performed RAF binding domain (RBD) assays followed by immunoblot analysis with KRAS, HRAS, NRAS, and KRAS^{G12D}-specific antibodies. As shown in Fig. 6E, the most significant increase in GTP levels were seen using the pan RAS and KRAS antibody. The KRAS^{G12D}-GTP levels were also slightly elevated in the absence of AGO2. HRAS-GTP levels were not elevated and we failed to detect NRAS. This suggests that although both forms of KRAS are activated in the absence of AGO2, the fraction of activated wild-type KRAS is higher than activated oncogenic KRAS.

Further, erlotinib treatment led to dramatic reduction in the wild type RAS-GTP levels in organoids with AGO2 loss (Fig. 6E-F). On the other hand, the levels of activated KRAS^{G12D} remained unchanged. Erlotinib treatment also reduced pERK levels reflecting inhibition of the EGFR/wild-type KRAS axis with minimal effect on oncogenic KRAS signaling that is predominant in AGO2-sufficient organoids. Note that pAKT levels did not change across all conditions. This provides experimental evidence for the hypothesis that AGO2 loss activates the EGFR/wild-type KRAS/ERK pathway.

(Fig. 6d) – In MEFs they show that AGO2 loss is associated with increased EGFR activity and RAS-GTP formation – is pERK but not pAKT increased as seen in mice with an AGO2

deficiency? Ectopic reexpression of AGO2 reversed these activities. The authors previously described an AGO2 mutant that cannot bind RAS, AGO2 K112A. Showing this mutant cannot reverse these activities would support the importance of AGO2 interaction with WT RAS in causing these signaling changes.

RESPONSE: We thank the reviewer for this important suggestion. To address this, we generated stable lines of AGO2^{-/-} MEFs expressing either vector, AGO2, or RAS binding deficient AGO2¹ (AGO2^{K112A/E114A}). As shown in Fig. 6H, EGFR activation in the AGO2 null MEFs could be rescued with the expression of both AGO2 and AGO2^{K112A/E114A} mutant. However, RAS-GTP levels and downstream ERK activation continued to be elevated in AGO2^{-/-} MEFs stably expressing the AGO2^{K112A/E114A} mutant. This provides evidence for the requirement of a direct interaction between AGO2 and KRAS to limit wild-type RAS activity and thereby its downstream signaling.

(Fig. 7a) – While the data do support the author's conclusion that EGF stimulation impairs AGO2 association with WT but not mutant KRAS, no data are presented to provide an explanation for this difference. Since the authors previously found that the AGO2-RAS interaction is independent of the nucleotide bound, the relative GTP-bound state of mutant versus WT KRAS should not be a factor.

RESPONSE: We agree with the reviewer that the nucleotide status of KRAS does not determine AGO2 binding, but, rather, the phosphorylation of AGO2 at Y393 determines its association with wild-type KRAS.

To explore this aspect further, we treated oncogenic KRAS-expressing cells with H₂O₂. Treatment with H₂O₂ inactivates tyrosine phosphatases by oxidation and leads to EGFR activation⁷. In both H358 and Mia PaCa-2 cells expressing oncogenic KRAS, the KRAS-AGO2 interaction could be disrupted following H₂O₂ treatment (Fig. 7G-H). This disruption was dependent on the Y393 phosphorylation site of AGO2, since mutant AGO2^{Y393F} remained recalcitrant to H₂O₂ treatment (Fig. 7I). This data mimics our earlier observation that AGO2^{Y393} phosphorylation by EGF stimulation disrupts the wild-type RAS interaction.

Together, these data suggest that the Y393 phosphorylation site in AGO2 determines binding of both wild-type and oncogenic forms of KRAS. While AGO2 phosphorylation in wild-type RAS expressing cells can be achieved by EGF stimulation, mutant RAS expressing cells requires sustained EGFR activation through inhibition of tyrosine phosphatases. We have summarized these data in Fig. 9.

(Fig. 7b) – The requirement for EGF stimulation to cause increased pERK levels in the KRAS mutant cancer lines is puzzling, since mutant KRAS alone, independent of upstream RTK signaling, should be capable of binding and activating RAF.

RESPONSE: The reviewer is correct, and this comment can be cleared up by the immunoblot shown below (Rebuttal Fig. 2). Cell lines expressing mutant KRAS indeed have higher levels of basal ERK signaling compared to cell lines with wild-type RAS under serum-starved conditions. EGF stimulation was carried out to maximize EGFR activation, thereby increasing the levels of AGO2 phosphorylation in both wild-type and mutant KRAS expressing cells.

Rebuttal Figure 2: Immunoblot analysis shows increased basal pERK levels in oncogenic KRAS expressing cells compared to wild-type RAS expressing cells.

(Fig. 8) – The ability of a elevG12C selective inhibitor to disrupt mutant KRAS-AGO1 interaction is confusing regarding what is known about AGO2-RAS interaction. The G12C inhibitor covalently modifies GDP-bound KRAS G12C, preventing formation of the GTP-bound state, leading to loss of effector binding. Since AGO2-RAS interaction is not nucleotide dependent, is this disruption due to loss of PI3K binding, or because the inhibitor-modified KRAS protein can no longer bind AGO2?

RESPONSE: We believe that the inhibitor-modified KRAS protein fails to bind AGO2. Specifically, the inhibitor is known to covalently attach to cysteine residues and makes contact at the Switch II pocket of GDP-bound RAS^{G12C}. This binding has been shown to interfere with nucleotide exchange because of reduced SOS1-KRAS^{G12C} interaction⁸. Given that SOS and AGO2 compete for binding to the Switch II domain to regulate RAS-GTP levels (Fig. 6I-K), both SOS and AGO2 fail to bind inhibitor-associated KRAS. Together, this suggests that RAS^{G12C} complexed with the inhibitor is unable to bind to both regulators (SOS1/AGO2) and effectors (PI3K) that interact with RAS via the Switch II domain.

(Fig. 9) – This model, while useful, is not necessary, since the concept is not complex and easily understandable by words in the text. However, some of the mechanistic issues that regulate AGO2-KRAS interaction, such as the role of EGFR signaling, would be better explained by a model figure. A model showing how EGFR activation can disrupt WT but not mutant KRAS association, and how an AGO2-RAS association alters RAS function, would be more informative.

RESPONSE: In the revised manuscript, we have summarized our observations regarding the regulation of the wild-type and mutant KRAS-AGO2 interaction through EGFR activation (Fig. 9). We have also updated the PDAC model to reflect our understanding of the changes in RNAi function and RAS activity due to AGO2 loss.

Reviewer #2 (Remarks to the Author); expert in AGO2 and RNAi machinery:

In this manuscript, Shankar et al. describe an unexpected role for AGO2 in KRAS^{G12D}-driven pancreatic cancer mouse model. They find that AGO2 is dispensable for the early stage of

cancer development but is required for the development of PDAC and metastasis. They demonstrate that AGO2 expression is required to block oncogene-induced senescence (OIS), which can be bypassed by abolishing p53 regulation. The authors propose that KRAS-AGO2 interaction is critical for this unconventional role of AGO2 in repressing OIS.

Overall, this study is solid and interesting. By mouse model studies, authors provide compelling evidence that AGO2 is required for PanINs to overcome OIS and progress to PDAC. The fact that AGO2 expression is elevated in human PDAC samples further supports the role of AGO2 plays in pancreatic cancer development.

However, the mechanistic part of this study is relatively weak. The same group has reported previously that KRAS-AGO2 interaction enhances cellular transformation based on cell line studies (Shankar et al, Cell Rep, 2016). Despite a fair amount of work, the current study does not bring much new to the underlying mechanism. The observation that phosphorylation of AGO2 by EGFR disrupts the interaction of WT KRAS, but not mutant KRAS, with AGO2, although by itself intriguing, does not contribute much towards the understanding of the most critical question: How does AGO2 overcome the OIS block? I would encourage the authors to address or at least discuss the following points/questions:

RESPONSE: We thank the reviewer for their overall positive and supportive comments. In the revised manuscript, we have provided key experimental evidence that addresses the concerns regarding the underlying mechanisms. Regarding the reviewer's main concern, to understand how AGO2 overcomes the OIS block, we analyzed microRNA expression and transcriptome profiles of pancreatic tissues obtained from $AGO2^{+/+};KRAS^{G12D};p48Cre$ and $AGO2^{fl/fl};KRAS^{G12D};p48Cre$ mice (Fig. 6A-B and Supplementary Figs. 10 and 11). As explained in the revised manuscript, AGO2 expression downregulates a set of microRNAs, including members of the let-7 family that regulate cell proliferation and Rb-regulated miR-29 and miR-30 families that have been shown to be associated with senescence. Further, transcriptome analysis shows a block in cell cycle progression in the PanINs of $AGO2^{fl/fl};KRAS^{G12D};p48Cre$ mice (Fig. 6B and Supplementary Fig. 11), supporting the OIS phenotype observed with AGO2 loss. More details are provided in the point-by-point responses below to specific critiques.

1. RAS-AGO2 interaction correlates with PDAC. It would greatly strengthen their conclusions if the authors can establish the causality in their mouse models.

RESPONSE: We agree with the reviewer that analyzing the RAS-AGO2 interaction would add strength to our conclusions. We now show analysis with stable cell lines expressing wild-type AGO2 or RAS binding-deficient AGO2 (K112A/E114A) to interrogate the specific effects of the KRAS-AGO2 interaction on RAS signaling. As shown in Fig. 6H, we find that binding of AGO2 to KRAS is required to limit downstream activation of ERK that is associated with OIS. As now stated in the Discussion section, our next step is to develop a mouse model with the KRAS binding-deficient AGO2 mutant to evaluate the role of the interaction *in vivo*, a study that is beyond the scope of a revised manuscript.

2. The inhibition of OIS is a result of modulating the function of AGO2, KRAS/KRAS-mut, or both? Specifically, the role of AGO2 in this process is microRNA dependent or independent?

RESPONSE: As described earlier, Rb-regulated microRNAs known to be elevated in senescence⁶ were found to be upregulated in pancreatic tissue with AGO2 loss (Fig. 6A). We believe that these microRNAs play a critical role in the development of OIS. Concomitant changes in RAS signaling were also observed in various models of AGO2 loss (Fig. 6C-K). Particularly, wild-type KRAS activation in the absence of AGO2 was evident in the pancreatic tissue, organoid, and MEF models (Fig. 6C and E-H). Given that changes in both RAS signaling and microRNA expression occur in the absence of AGO2, it is difficult to discern the primary trigger for the OIS.

How is the function of KRAS/KRAS-mut impacted by the AGO2 association?

RESPONSE: In Fig. 6, we have now detailed the mechanisms that explain the AGO2 interaction with both the wild-type and oncogenic forms of KRAS. Activation of wild-type RAS in the *AGO2*^{-/-} MEFs (Fig. 6G) suggests that AGO2 limits wild-type RAS activation through its interaction with RAS (Fig. 6H). This regulation of wild-type RAS activity can be attributed to the ability of AGO2 to compete with nucleotide exchange factors like SOS (Fig. 6I-K) that bind to the same region in RAS. AGO2 loss in the organoid model also leads to a moderate increase in the activated forms of oncogenic KRAS. Together, AGO2 association with both the wild-type and mutant KRAS restricts nucleotide exchange on KRAS (summarized in Fig. 9).

Transcriptome analyses by RNA-Seq as well as the profiling of miRNAs by miRNA-Seq should provide insights.

RESPONSE: In the revised manuscript, we have now included our analyses of the microRNA expression and transcriptomic profiles that provide an in-depth understanding of the mechanisms that explain the phenotype of AGO2 loss (Fig. 6A-B and Supplementary Figs. 10-11). As noted above, we find changes in expression of microRNAs that regulate cell proliferation (*let-7*) and senescence (*miR-29* and *miR-30* family members). These changes in microRNA levels are accompanied by the downregulation of genes involved in E2F signaling and G2/M cell cycle progression.

Reviewer #3 (Remarks to the Author); expert in oncogene-induced senescence:

Manuscript: #NCOMMS-19-25257-T

Title: An essential role for argonaute 2 in EGFR-KRAS signaling in pancreatic cancer development

Authors: Shankar et al.

In the manuscript "An essential role for argonaute 2 in EGFR-KRAS signaling in pancreatic cancer development" (#NCOMMS-19-25257-T), Shankar and colleagues delineate a complex signal transduction pathway whereby epidermal growth factor receptor (EGFR) signaling provokes the dissociation of a supramolecular complex between wild-type (but not mutant) KRAS and argonaute 2 (AGO2), which is relevant for pancreatic oncogenesis driven by

KRASG12D, as the latter occurs in an initial EGFR-dependent phase, and a subsequent EGFR-independent phase.

The findings by Shankar et al. are novel and the study is globally well performed and in line with the scope of Nature Communications. The manuscript is well written and figures are clear, although not particularly well organized (and in some instances contains excessive/inaccurate statements). The technological portfolio employed by the authors to demonstrate their hypotheses is not particularly variegated (largely, imaging often not combined with quantitative analysis, immunoblot and co-immunoprecipitation), but globally serves the purpose. In summary, I believe this work is a good candidate for publication in Nature Communications, once the following points have been properly addressed.

RESPONSE: We thank the reviewer for their comments and enthusiasm for this study. In the revised manuscript, we have included a wider variety of approaches. The use of biochemical assays with purified proteins, microRNA expression profiles, and transcriptomic analysis have provided important mechanistic insights that are highlighted in Figure 6.

Major issues:

1. Little has been done about the cellular mechanisms whereby the AGO2-KRAS complex impact on oncogenesis and tumor progression. The data are strikingly similar to work from Eileen White demonstrating that defects in Atg7 favor early oncogenesis in the lung and yet prevent progression to advanced disease stage. Given the results with serum starvation and the well described role EGFR and KRAS in autophagy inhibition, investigating the impact of autophagy would be a great addition here.

RESPONSE: The reviewer raises an interesting aspect of OIS that we explored in our PDAC model with AGO2 loss. As seen in the Rebuttal Figure 3 below, AGO2 ablation dramatically reduces the autophagic flux in the PanIN lesions that fail to progress to PDAC. Given that autophagy is critical for KRAS-driven cellular transformation in both pancreatic⁹ and lung cancer¹⁰ progression, the role for AGO2 to regulate this pathway is

[REDACTED]

interesting and needs further investigation.

2. Along similar lines, it is surprising that the authors did not investigate whether the control of lesions undergoing senescence in the absence of AGO2 is mediated by the immune system.

Senescence surveillance by NK cells is well established and could play a major role in this model.

RESPONSE: We thank the reviewer for this suggestion, and, as predicted by the reviewer, NK cells do indeed play a major role in this model. To address this, we have now compared the immunogenic profile of senescent PanIN lesions with AGO2 loss to PanIN/PDAC lesions expressing AGO2. As shown in Fig. 2E, both CD8⁺ T lymphocytes and natural killer (NK) cells were significantly enriched upon AGO2 loss. It is interesting to note that both of these cells share properties and interact to elicit cytolytic outcomes¹¹.

Given that NK cells are essential for immunosurveillance of senescent cells^{12,13}, we compared NK cell populations in the PanINs within the pancreas of the two genotypes. As this reviewer predicted, there was a significant increase in the NK cell population specifically in pancreatic tissues with AGO2 loss. The increase in NK cell population was observed both at the periphery and, more importantly, in close proximity to the senescent cells (Fig. 2F-H). It remains possible that over time, these senescent cells are cleared by NK cells through perforin release by exocytosis¹⁴.

3. Do similar observations apply to other types of cancer driven by KRAS mutations, like lung carcinoma? Understanding the extension of the phenomenon would be important

RESPONSE: Preliminary data suggests that AGO2 is critical for progression of benign adenomas to grade 4 adenocarcinomas in the KRAS-driven lung model (Rebuttal Figure 4). This phenotype is similar to the requirement for AGO2 in the pancreatic model, yet it is important to note that in the lung model, AGO2 dependency is observed even in the absence of p53, suggesting a contextual role for AGO2 in KRAS-driven cancer progression. We are currently exploring this model in-depth, and it will be the subject of a follow-up manuscript dedicated to the role of the RAS-AGO2 axis in lung cancer.

[REDACTED]

4. A transcriptional comparison of AGO2 competent and incompetent lesions is needed to obtain additional insights into the underlying mechanisms

RESPONSE: We thank the reviewer for this suggestion which has added important information to our study. In the revised manuscript, we have now included our analyses of the microRNA expression and transcriptomic profiles that provide an in-depth understanding of the mechanisms that explain the phenotype of AGO2 loss (Fig. 6A-B and Supplementary Figs. 10-11).

Briefly, our mouse model reveals a central role for both let-7 microRNAs and the Rb-dependent miR-29 and miR-30 families of microRNAs to control cellular senescence through AGO2, which is overcome upon p53 loss (Fig. 6A and Supplementary Fig. 10). Additionally, transcriptomic profiling shows a lack of cell cycle progression in AGO2-deficient lesions accounting for the block in PDAC progression (Fig. 6B and Supplementary Fig. 11).

5. Fig. 6A,B. Total EGFR levels change considerably in immunoblot, not in IHC. How do the authors explain this? In general, many immunoblots require densitometry and quantitation.

RESPONSE: In the revised manuscript, we have included the quantitation of the blots from the pancreatic tissue. As can be seen in Supplementary Fig. 12B, the interpretation of the data has not changed. We further performed quantitative analysis of EGFR IHC staining with the same antibody used in the immunoblot analysis. As shown in Supplementary Fig. 13B, there is no significant difference in EGFR expression between the two genotypes analyzed.

6. Fig 7G is missing, which prevents evaluation of the corresponding text.

RESPONSE: We sincerely apologize for this error and have now included this figure in the revised manuscript.

Minor issues:

1. Some statements and data interpretation are inaccurate or excessive and need to be corrected or toned down. As a few examples: line 83 “closely mimics” is excessive, given the exclusivity of KRAS and EGFR mutations in human but the need for EGFR signaling in the model; line 112 “faithfully mimicking”, same problem; line 386, this is not true, it can reflect a distant binding and conformational change.

RESPONSE: In the revised manuscript, we have made the changes that the reviewer has suggested. We have also corrected our interpretation of RAS and AGO2 staining, limiting our statements of co-localization and interaction to only the PLA based assays.

2. Fig. 5E, the authors points to increase levels of activated KRAS between PanINs and PDACs, which is not completely supported by non-quantitative imaging

RESPONSE: We have now quantitated the RAS (Supplementary Fig. 7F) and AGO2 (Supplementary Fig. 7G) levels. The data validates our previous statement that RAS expression is increased in PDAC compared to PanINs.

3. Figure 3A. The authors conclude that p53 knockout can abolish the survival advantage obtained by AGO2 knockout (Fig. 1F), but the two scenarios are difficult to compare as median survival is completely different. This should be commented upon: the disease in the absence of p53 is a completely different one

RESPONSE: In the revised manuscript we have added the comment below that the reviewer suggests.

“Therefore, the mouse model with p53 loss with aggressive disease represents a different etiology for PDAC progression and limits our understanding of the requirement of AGO2 in advanced disease states.”

4. Figure 3C. AGO2 positive islands are no longer visible in the Ago2 KO setting, at odds with Fig 1B.

RESPONSE: In the revised manuscript, we have added a representative image of AGO2 IHC in pancreatic tissue of $AGO2^{+/+};KRAS^{G12D};p53^{fl/+};p48Cre$ which shows increased AGO2 staining in the pancreatic islets (Supplementary Fig. 9A). The staining shown in Fig. 3C is limited to the PDAC regions with no islets within this genotype.

5. Figure 4B, the authors may want to explore TCGA data in support of their hypothesis.

RESPONSE: We thank the reviewer for this suggestion. In the revised manuscript, we have included data that suggests AGO2 mRNA levels are significantly elevated in PDAC compared to normal tissues (Supplementary Fig. 7A).

6. It would have been preferable to have composed figures rather than sparse panels.

RESPONSE: We postulate that the reviewer was referring to Figures 5 and 7 that were split across multiple pages. This was done to provide images with higher magnification for the peer review process. As the reviewer suggests, the final images can be combined at the production stage for the journal.

1. Shankar, S., *et al.* KRAS Engages AGO2 to Enhance Cellular Transformation. *Cell Rep* 14, 1448-1461 (2016).

2. Tsherniak, A., *et al.* Defining a Cancer Dependency Map. *Cell* **170**, 564-576 e516 (2017).
3. Omerovic, J., Hammond, D.E., Clague, M.J. & Prior, I.A. Ras isoform abundance and signalling in human cancer cell lines. *Oncogene* **27**, 2754-2762 (2008).
4. Leroy, B., *et al.* Analysis of TP53 mutation status in human cancer cell lines: a reassessment. *Hum Mutat* **35**, 756-765 (2014).
5. Martinez, I., Cazalla, D., Almstead, L.L., Steitz, J.A. & DiMaio, D. miR-29 and miR-30 regulate B-Myb expression during cellular senescence. *Proc Natl Acad Sci U S A* **108**, 522-527 (2011).
6. Suh, N. MicroRNA controls of cellular senescence. *BMB Rep* **51**, 493-499 (2018).
7. Tan, X., Lambert, P.F., Rapraeger, A.C. & Anderson, R.A. Stress-Induced EGFR Trafficking: Mechanisms, Functions, and Therapeutic Implications. *Trends Cell Biol* **26**, 352-366 (2016).
8. Lito, P., Solomon, M., Li, L.S., Hansen, R. & Rosen, N. Allele-specific inhibitors inactivate mutant KRAS G12C by a trapping mechanism. *Science* **351**, 604-608 (2016).
9. Yang, A., *et al.* Autophagy Sustains Pancreatic Cancer Growth through Both Cell-Autonomous and Nonautonomous Mechanisms. *Cancer Discov* **8**, 276-287 (2018).
10. Guo, J.Y., *et al.* Autophagy suppresses progression of K-ras-induced lung tumors to oncocytomas and maintains lipid homeostasis. *Genes Dev* **27**, 1447-1461 (2013).
11. Sun, J.C. & Lanier, L.L. NK cell development, homeostasis and function: parallels with CD8(+) T cells. *Nat Rev Immunol* **11**, 645-657 (2011).
12. Krizhanovsky, V., *et al.* Implications of cellular senescence in tissue damage response, tumor suppression, and stem cell biology. *Cold Spring Harb Symp Quant Biol* **73**, 513-522 (2008).
13. Krizhanovsky, V., *et al.* Senescence of activated stellate cells limits liver fibrosis. *Cell* **134**, 657-667 (2008).
14. Sagiv, A., *et al.* Granule exocytosis mediates immune surveillance of senescent cells. *Oncogene* **32**, 1971-1977 (2013).

REVIEWERS' COMMENTS:

Reviewer #1 (Remarks to the Author):

The authors have done a nice job with responding to concerns raised. The concerns have been addressed, in part with clarification and text revisions, in part with additional experimental data. This revised manuscript is now acceptable for publication with no further revisions needed.

Reviewer #2 (Remarks to the Author):

The authors did an impressive job of addressing reviewers' concerns. The manuscript is much improved.

I find it odd that AGO2 depletion led to up-regulation of a subset of miRNAs, given that AGOs were shown to stabilize miRNAs in general. While future studies are clearly required to establish the mechanism(s) by which AGO2 contributes to OIS regulation, this study by itself is interesting and in my opinion, should be published.

Reviewer #3 (Remarks to the Author):

Manuscript: #NCOMMS-19-25257A

Title: An essential role for argonaute 2 in EGFR-KRAS signaling in pancreatic cancer development

Authors: Shankar et al.

In the original version of the manuscript " An essential role for argonaute 2 in EGFR-KRAS signaling in pancreatic cancer development" (#NCOMMS-19-25257-T), Shankar and colleagues delineated a complex signal transduction pathway whereby epidermal growth factor receptor (EGFR) signaling provokes the dissociation of a supramolecular complex between wild-type (but not mutant) KRAS and argonaute 2 (AGO2), which is relevant for pancreatic oncogenesis driven by KRASG12D, as the latter occurs in an initial EGFR-dependent phase, and a subsequent EGFR-independent phase.

When I first evaluated the article, I raised a number of points that were for the most part addressed by the authors. However, there are some outstanding issues that still require attention before publication

Major issues:

1. IMPORTANT: the authors confirmed my original suspicion that an autophagy defect is involved in their observations, but dismissed this as part of the rebuttal letter. Although the legend to F3 of the rebuttal letter is not complete (as it is unclear what each lane represents), the deletion of AGO2 does not simply "impair autophagic flux" (which the authors probably did not measure, unless these samples came from mice optionally starved in the optional presence of a lysosomal inhibitor before sacrifice) but fully abolish LC3 expression (no LC3-I, which obviously prevents LC3-II lipidation). The impact of autophagy in their model must be further validated by crossing AGO2^{fl/fl} mice with ATG7^{fl/fl} mice, and both the latter and their double floxed AGO2^{fl/fl}/ATG7^{fl/fl} progeny with KRASG12Dp48Cre mice to compare oncogenesis and tumor progression. As it stands, all the observations from Shankar and colleagues may reflect autophagy inhibition (see PMIDs 23824538, 25673642, 23965987, 24445999 for the experimental framework to my comment), and this needs to be elucidated.

2. In further support of the above, both miR-29 and miR-30 (which now the authors identify as part of the miRNAs upregulated by AGO2 depletion) have previously been involved in pancreatic

autophagy (PMID 27626694) or gastric autophagy (PMID 26209295) regulation. Moreover, autophagy is known to be required for the establishment and maintenance of OIS (see PMID 26524528)

3. The authors measured recruitment of NK cells, which is an observational support to the notion that NK cells may control (at least partially) cells undergoing OIS, and commented in the rebuttal letter that "It remains possible that over time, these senescent cells are cleared by NK cells through perforin release by exocytosis". Mechanistic studies in the presence of anti-NKGD2 or anti-NK1.1 antibodies (both of which deplete NK cells and some T cells, depending on the mouse background), versus antiCD4+antiCD8 antibodies (which deplete T cells only) would be importantly to clarify whether NK cells actually control OIS (and hence tumorigenesis is accelerated) or are simple bystanders (and hence tumorigenesis is not affected)

END

NCOMMS-19-25257A: Response to Reviewers' Comments

Reviewer #1 (Remarks to the Author):

The authors have done a nice job with responding to concerns raised. The concerns have been addressed, in part with clarification and text revisions, in part with additional experimental data. This revised manuscript is now acceptable for publication with no further revisions needed.

RESPONSE: We thank the reviewer and appreciate the positive comments on the revised manuscript.

Reviewer #2 (Remarks to the Author):

The authors did an impressive job of addressing reviewers' concerns. The manuscript is much improved.

I find it odd that AGO2 depletion led to up-regulation of a subset of miRNAs, given that AGOs were shown to stabilize miRNAs in general. While future studies are clearly required to establish the mechanism(s) by which AGO2 contributes to OIS regulation, this study by itself is interesting and in my opinion, should be published.

RESPONSE: We thank the reviewer for sharing our enthusiasm for the revised manuscript. We also agree with the reviewer that further studies are required to understand how AGO2 loss results in the upregulation of specific microRNAs leading to OIS.

Reviewer #3 (Remarks to the Author):

Manuscript: #NCOMMS-19-25257A

Title: An essential role for argonaute 2 in EGFR-KRAS signaling in pancreatic cancer development

Authors: Shankar et al.

In the original version of the manuscript " An essential role for argonaute 2 in EGFR-KRAS signaling in pancreatic cancer development" (#NCOMMS-19-25257-T), Shankar and colleagues delineated a complex signal transduction pathway whereby epidermal growth factor receptor (EGFR) signaling provokes the dissociation of a supramolecular complex between wild-type (but not mutant) KRAS and argonaute 2 (AGO2), which is relevant for pancreatic oncogenesis driven by KRASG12D, as the latter occurs in an initial EGFR-dependent phase, and a subsequent EGFR-independent phase.

When I first evaluated the article, I raised a number of points that were for the most part

addressed by the authors. However, there are some outstanding issues that still require attention before publication

Major issues:

1. **IMPORTANT:** the authors confirmed my original suspicion that an autophagy defect is involved in their observations, but dismissed this as part of the rebuttal letter. Although the legend to F3 of the rebuttal letter is not complete (as it is unclear what each lane represents), the deletion of AGO2 does not simply “impair autophagic flux” (which the authors probably did not measure, unless these samples came from mice optionally starved in the optional presence of a lysosomal inhibitor before sacrifice) but fully abolish LC3 expression (no LC3-I, which obviously prevents LC3-II lipidation). The impact of autophagy in their model must be further validated by crossing AGO2fl/fl mice with ATG7fl/fl mice, and both the latter and their double floxed AGO2fl/flATG7fl/fl progeny with KRASG12Dp48Cre mice to compare oncogenesis and tumor progression. As it stands, all the observations from Shankar and colleagues may reflect autophagy inhibition (see PMIDs 23824538, 25673642, 23965987, 24445999 for the experimental framework to my comment), and this needs to be elucidated.

RESPONSE: We would like to thank the reviewer again for the excellent point regarding autophagy in this model. We included this exclusively in the rebuttal letter to highlight our intent to pursue these studies further in a future manuscript with no intent to dismiss. We also believe that autophagy inhibition may lead to OIS in this model. The animal experiments suggested by the reviewer are also valid and will help elucidate the role of AGO2 in autophagy. However, this first study with AGO2 loss is focused on its role in KRAS-driven PDAC and a deeper understanding of the KRAS-AGO2 interaction and its regulation by EGFR. To address the reviewer’s concern, we have suggested that AGO2 may regulate autophagy in the Discussion section of this manuscript.

2. In further support of the above, both miR-29 and miR-30 (which now the authors identify as part of the miRNAs upregulated by AGO2 depletion) have previously been involved in pancreatic autophagy (PMID 27626694) or gastric autophagy (PMID 26209295) regulation. Moreover, autophagy is known to be required for the establishment and maintenance of OIS (see PMID 26524528)

RESPONSE: We thank the reviewer for raising this important point. We have now included these references in the Discussion section that provide evidence for the role of miR-29/miR-30 family microRNAs in autophagy to explain the effects of AGO2 loss.

3. The authors measured recruitment of NK cells, which is an observational support to the notion that NK cells may control (at least partially) cells undergoing OIS, and commented in the rebuttal letter that “It remains possible that over time, these senescent cells are cleared by NK cells through perforin release by exocytosis”. Mechanistic studies in the presence of anti-NKGD2 or anti-NK1.1 antibodies (both of which deplete NK cells and some T cells, depending on the mouse background), versus antiCD4+antiCD8 antibodies (which deplete T cells only) would be importantly to clarify whether NK cells actually control OIS (and hence tumorigenesis is accelerated) or are simple bystanders (and hence tumorigenesis is not affected)

RESPONSE: We thank the reviewer for raising the question of NK cell recruitment in the

PDAC model. We think it provides great value to the community and likely explains why the OIS phenotype is lost over an extended period of time. Experimental work along the lines that the reviewer suggests will be carried out for a future study to address this notion.